# Climate model projections from the Scenario Model Intercomparison Project (ScenarioMIP) of CMIP6

Claudia Tebaldi[1], Kevin Debeire[2,3], Veronika Eyring[2,4], Erich Fischer[5], John Fyfe[6], Pierre Friedlingstein[7,8], Reto Knutti[5], Jason Lowe[9,10], Brian O'Neill[11], Benjamin Sanderson[12], Detlef van Vuuren[13], Keywan Riahi[14], Malte Meinshausen[15], Zebedee Nicholls[15], Katarzyna B. Tokarska[5], George Hurtt[16], Elmar Kriegler[17], Jean-Francois Lamarque[18], Gerald Meehl[18], Richard Moss[1], Susanne E. Bauer[19], Olivier Boucher[20], Victor Brovkin[21], Young-Hwa Byun[22], Martin Dix[23], Silvio Gualdi[24], Huan Guo[25], Jasmin G. John[25], Slava Kharin[6], Tsuyoshi Koshiro[26], Libin Ma[27], Dirk Olivié[28], Swapna Panickal[29], Fangli Qiao[30], Xinyao Rong[31], Nan Rosenbloom[18],  Martin Schupfner[32], Roland Séférian[33], Alistair Sellar[9], Tido Semmler[34], Xiaoying Shi[35], Zhenya Song[30], Christian Steger[36], Ronald Stouffer[37], Neil Swart[6], Kaoru Tachiiri[38], Qi Tang[39], Hiroaki Tatebe[38], Aurore Voldoire[33], Evgeny Volodin[40], Klaus Wyser[41], Xiaoge Xin[42], Shuting Yang[43], Yongqiang Yu[44], Tilo Ziehn[23].

[1]Joint Global Change Research Institute (JGCRI), Pacific Northwest National Laboratory, College Park, MD, USA.
[2]Deutsches Zentrum für Luft- und Raumfahrt (DLR), Institut für Physik der Atmosphäre, Oberpfaffenhofen, Germany.
[3]Deutsches Zentrum für Luft- und Raumfahrt (DLR), Institut für Datenwissenschaften, Jena, Germany.
[4]University of Bremen, Institute of Environmental Physics (IUP), Bremen, Germany.
[5]ETH Zurich, Institute for Atmospheric and Climate Science, Zurich, Switzerland.
[6]Canadian Centre for Climate Modelling and Analysis, Environment and Climate Change Canada, Victoria, BC, Canada.
[7]College of Engineering, Mathematics and Physical Sciences, University of Exeter, Exeter, EX4 4QE, United Kingdom.
[8]LMD/IPSL, ENS, PSL Université, Ècole Polytechnique, Institut Polytechnique de Paris, Sorbonne Université, CNRS, Paris, France.
[9]Met Office Hadley Center, Exeter, UK.
[10]Priestley International Center for Climate, School of Earth and Environment, University of Leeds, Leeds, UK.
[11]Josef Korbel School of International Studies, University of Denver, Denver, USA.
[12]CNRS/Centre Européen de Recherche et de Formation Avancée en Calcul Scientifique (CERFACS), Toulouse, France.
[13]PBL Netherlands Environmental Assessment Agency and Faculty of Geosciences, Utrecht University, Utrecht, The Nederlands.
[14]International Institute for Applied Systems Analysis, Laxenburg, Austria.
[15]Climate & Energy College, School of Earth Sciences, University of Melbourne, Australia.
[16]Department of Geographical Sciences, University of Maryland, College Park, MD, USA.
[17]Potsdam Institute for Climate Impact Research (PIK), Potsdam, Germany.
[18]Climate and Global Dynamics Laboratory, National Center for Atmospheric Research, Boulder, CO, USA.
[19]NASA Goddard Institute for Space Studies, New York, NY, USA.
[20]Institut Pierre-Simon Laplace, Sorbonne Université / CNRS, Paris, France.

[21]Max Planck Institute for Meteorology, Hamburg, Germany; also: Center for Earth System Research and Sustainability, University of Hamburg, Germany.

[22]National Institute of Meteorological Sciences / Korea Meteorological Administration, South Korea.

[23]CSIRO Oceans and Atmosphere, Aspendale, Victoria, Australia.

[24]Centro Euro-Mediterraneo sui Cambiamenti Climatici (CMCC), Italy.

[25]NOAA/OAR/Geophysical Fluid Dynamics Laboratory, Princeton, NJ, USA.

[26]Meteorological Research Institute, Tsukuba, Japan.

[27]Earth System Modeling Center, Nanjing University of Information Science and Technology, Jiangsu, China.

[28]Norwegian Meteorological Institute, Oslo, Norway.

[29] Indian Institute of Tropical Meteorology, Pune, India.

[30]First Institute of Oceanography (FIO), Ministry of Natural Resources (MNR), Qingdao, China.

[31]State Key Laboratory of Severe Weather, Chinese Academy of Meteorological Sciences, Beijing, China.

[32]Deutsches Klimarechenzentrum, Hamburg, Germany.

[33]CNRM, Université de Toulouse, Météo-France, CNRS, Toulouse, France.

[34]Alfred Wegener Institute, Helmholtz Centre for Polar and Marine Research, Bremerhaven, Germany.

[35]Oak Ridge National Laboratory, Oak Ridge, TN, USA.

[36]Deutscher Wetterdienst, Offenbach, Germany.

[37]University of Arizona, Tucson, AR, USA.

[38]Research Institute for Global Change (RIGC), Japan Agency for Marine-Earth Science and Technology (JAMSTEC), Yokohama, Japan.

[39]Lawrence Livermore National Laboratory, Livermore, CA, USA.

[40]Institute of Numerical Mathematics, Moscow, Russian Federation.

[41]Swedish Meteorological and Hydrological Institute, Norrkoeping, Sweden.

[42]Beijing Climate Center, China Meteorological Administration, Beijing, China.

[43]Danish Meteorological Institute, Copenhagen, Denmark.

[44]LASG, Institute of Atmospheric Physics, Chinese Academy of Sciences, Beijing, China.

*Correspondence to* Claudia Tebaldi (claudia.tebaldi@pnnl.gov)


**Abstract.** The Scenario Model Intercomparison Project (ScenarioMIP) defines and coordinates the main set of future climate projections, based on concentration driven simulations, within the Coupled Model Intercomparison Project Phase 6 (CMIP6). This paper presents a range of its outcomes by synthesizing results from the participating global coupled Earth system models. We limit our scope to the analysis of strictly geophysical outcomes: mainly global averages and spatial patterns of change for surface air temperature and precipitation. We also compare CMIP6 projections to CMIP5 results, especially for those scenarios that were designed to provide continuity across the CMIP phases, at the same time highlighting important differences in forcing composition, as well as in results. The range of future temperature and precipitation changes by the end of the century (2081-2100) encompassing the Tier 1 experiments (SSP1-2.6, SSP2-4.5, SSP3-7.0 and SSP5-8.5) and SSP1-1.9 spans a larger range of outcomes compared to CMIP5, due to higher warming (by close to 1.5°C) reached at the upper end of the 5-95% envelope of the highest scenario, SSP5-8.5. This is due to both the wider range of radiative forcing that the new scenarios cover and to higher climate sensitivities in some of the new models compared to their CMIP5 predecessors. Spatial patterns of change for temperature and precipitation averaged over models and scenarios have familiar features, and an analysis of their variations confirms model structural differences to be the dominant source of uncertainty. Models also differ with respect to the size and evolution of internal variability as measured by individual models' initial condition ensembles' spread, according to a set of initial condition ensemble simulations available under SSP3-7.0. These experiments suggest a tendency for internal variability to decrease along the course of the century in this scenario, a result that will benefit from further analysis over a larger set of models. Benefits of mitigation, all else being equal in terms of societal drivers, appear clearly when comparing scenarios developed under the same SSP, but to which different degrees of mitigation have been applied. It is also found that a mild overshoot in temperature of a few decades in mid-century, as represented in SSP5-3.4OS, does not affect the end outcome of temperature and precipitation changes by 2100, which return to the same level as those reached by the gradually increasing SSP4-3.4 (not erasing the possibility, however, that other aspects of the system may not be as easily reversible). Central estimates of the time at which the ensemble means of the different scenarios reach a given warming level might be biased by the inclusion of models that have shown faster warming in the historical period than the observed. Those estimates show all scenarios reaching 1.5°C of warming compared to the 1850-1900 baseline in the second half of the current decade, with the time span between slow and fast warming covering between 20 and 27 years from present. 2°C of warming is reached as early

as 2039 by the ensemble mean under SSP5-8.5, but as late as the mid-'60s under SSP1-2.6. The highest warming level considered, 5°C, is reached only by the ensemble mean under SSP5-8.5, and not until the mid-'90s.


## 1. Introduction

Multi-model climate projections represent an essential source of information for mitigation and adaptation decisions. O'Neill et al. (2016) describe the origin, rationale and details of the experimental design for the Scenario Model Intercomparison Project (ScenarioMIP) for the Coupled Model
Intercomparison Project Phase 6 (CMIP6, Eyring et al, 2016). The experiments produce projections for a set of eight new 21st century scenarios based on the Shared Socio-economic Pathways (SSPs) and developed by a number of Integrated Assessment Models (IAMs). Extensions beyond 2100 based on idealized pathways of anthropogenic forcings are also included (formalized in their protocol by Meinshausen et al. (2020)), together with the request for a large initial condition ensemble under one of
the 21st century scenarios. Two of the scenarios are concentration overshoot (peak and decline) trajectories, while the majority follow a traditional increasing or stabilizing trajectory.
The new scenarios are the result of an intense research phase that produced a new systematic scenario approach, the SSP-RCP framework (van Vuuren et al., 2013), which relates the newer socio-economic scenarios to the Representative Concentration Pathways first adopted in CMIP5 (Moss et al., 2010;
Taylor et al., 2012). New qualitative narratives and future pathways of socio-economic drivers (population, technology and GDP) were developed according to two dimensions relevant to the climate change problem, i.e., by positioning individual pathways as each representing a combination of low, medium or high degrees of challenge to adaptation and to mitigation (O'Neill et al., 2013). Five such pathways (SSP1 through SSP5) were developed. These were in turn used by IAMs to produce scenarios
of anthropogenic emissions and land use (Riahi et al., 2017) consistent with the qualitative narratives and quantitative elements of each SSP. In addition to these baseline scenarios (i.e., scenarios that assume no explicit mitigation policies beyond those in place at the time the scenarios were created, prior to the Paris Agreement), a number of additional emissions and land use scenarios were produced that included mitigation policies (Kriegler et al., 2014) that achieved a range of radiative forcing targets for
the end of the century. Thus, on the basis of a given SSP multiple levels of radiative forcings are achievable, given more or less stringent mitigation. Among this large set of scenarios, the ScenarioMIP design chose a subset to be run by global climate and Earth System Models (ESMs) in concentration driven mode. Some were chosen specifically to provide continuity with the RCPs: SSP1-2.6, SSP2-4.5, SSP4-6.0 and SSP5-8.5, where 2.6 to 8.5 stands for the stratospheric adjusted radiative forcing in Wm$^{-2}$
by the end of the 21st century as estimated by the IAMs. Additional trajectories were also chosen to fill in gaps in the previous scenario set for both baseline and mitigation scenarios (SSP5-3.4; SSP3-7.0). Yet another was chosen to address new policy objectives (SSP1-1.9, designed to meet the 1.5°C target at the end of the century). The request of prioritizing initial condition ensemble members for only one of the scenarios (SSP3-7.0) was aimed at gathering sizable ensembles (10 members or more) from
various modelling centers. This was decided in recognition of the important role of internal variability in contributing to future changes, whose exploration is facilitated by initial condition ensembles (Deser et al., 2020; Santer et al., 2019). It was also recognized that the spread in aerosol scenarios in the four

RCPs used in CMIP5 was too narrow, as all assumed a large reduction in atmospheric aerosol emissions (Moss et al. 2010, Stouffer et al., 2017). The new SSP-based scenarios better address this uncertainty by
sampling a larger range of aerosols pathways consistent with the corresponding GHG emissions (Riahi et al. 2017). Scenario experiments were enabled by another community effort, input4mip: Based on the IAM's emission trajectories, and after harmonization of those to historical emission levels (Gidden et al., 2019), a community effort took place to translate those emission time series and to amend them with additional input fields for use by ESMs. These range from providing land-use patterns
(https://doi.org/10.22033/ESGF/input4MIPs.1127), gridded aerosol emission fields (Hoesly et al, 2018), stratospheric aerosols (Thomason et al., 2018), solar irradiance time series (Mattes et al., 2017), greenhouse gas concentrations (Meinshausen et al., 2020), as well as ozone fields (https://doi.org/10.22033/ESGF/input4MIPs.1115).

Given the multi-model focus of CMIP and the overview purpose of this paper, the results reported here
aim at giving a broadscale representation of ensemble results (mean and ranges, or other measures of variability). The ScenarioMIP design responded to many complex objectives and science questions, among which a high priority was the need to lay the foundation for integrated research across the geophysical, mitigation, impacts, adaptation and vulnerability research communities (O'Neill et al., 2020).  The focus of this paper is to provide physical climate context for these more detailed analyses.
Other Model Intercomparison Projects within CMIP6 have prescribed experiments that complement the ScenarioMIP design to address questions about the effects of small radiative forcing differences, specific (and often local) forcings like from land-use and short-lived climate forcers (SLCFs), the differential effects of emission versus concentration driven experiments testing the strength of the carbon cycle (Arora et al., 2020), and the effectiveness of emergent constraints in reshaping the
uncertainty ranges of the new multi-model ensemble (Nijsse et al., 2020; Tokarska et al., 2020). They are the Land Use MIP (LUMIP, Lawrence et al., 2016), the Aerosol Chemistry MIP (AerChemMIP, Collins et al., 2017), the Coupled Climate-Carbon Cycle MIP (C4MIP, Jones et al., 2016), the Geoengineering MIP (GeoMIP, Kravitz et al., 2015) and the Carbon Dioxide Removal MIP (CDRMIP, Keller et al., 2018).
In this study, we focus the analysis on the future evolution of average temperatures and precipitation. We address questions regarding the strength of the signal under the different CMIP6 scenarios and compared to similar CMIP5 scenarios; the identification of the time of separation between the temperature trajectories under the different scenarios, and the time at which they cross global warming thresholds. We also analyze spatial patterns of change addressing questions of robustness between the
CMIP5 and CMIP6 multi-model ensembles, and within the CMIP6 ensemble among models and scenarios.

**2. ScenarioMIP experiments and participating models**
As described in detail in O'Neill et al. (2016) and summarized in the matrix display of Fig. A1 in the Appendix, the ScenarioMIP design consists of the following concentration-driven scenario experiments, subdivided into two tiers to guide prioritization of computing resources. Tier 1 consists of four 21st century scenarios.  Three of them provide continuity with CMIP5 RCPs by targeting a similar level of aggregated radiative forcing (but we highlight important differences in the coming discussion): *SSP1-*

*2.6, SSP2-4.5 and SSP5-8.5.* An additional scenario, *SSP3-7.0*, fills a gap in the medium to high end of the range of future forcing pathways with a new baseline scenario, assuming no additional mitigation beyond what is currently in force. The same scenario also prescribes larger SLCFs concentrations and land-use changes compared to the other trajectories.

Only Tier 1, which can be satisfied by one realization per model, is required for participation in ScenarioMIP.
Tier 2 completes the design by adding

- *SSP1-1.9,* informing the Paris Agreement target of 1.5°C above pre-industrial;

- *SSP4-3.4,* a gap-filling mitigation scenario;

- *SSP4-6.0,* an update of the CMIP5-era RCP6.0;

- *SSP5-3.4OS (overshoot),* that tests the efficacy of an accelerated uptake of mitigation measures after a delay in curbing emissions until 2040: the scenario tracks SSP5-8.5 until that date, then decreases to the same radiative forcing of SSP4-3.4 by 2100;

- *three extensions to 2300*, two of them continuing on from SSP1-2.6 and SSP5-8.5 and one

extending the SSP5-3.4 overshoot pathway towards the lower radiative forcing level of 2.6 $Wm^{-2}$, to inform the analysis of long-memory processes, like ice-sheet melting and corresponding sea level rise.

- *nine additional initial condition ensemble members under SSP3-7.0* to explore internal variability and signal to noise characteristics of the different participating models.


A list of the participating models, with references for documentation and data, is shown in Table A1 in the Appendix. Table A2 lists the CMIP5 models used in the comparisons.

**3. Results**
For the results shown in this section we extracted monthly mean near-surface air temperature (TAS) and precipitation (PR) from the models listed in Table A1 and A2 (for CMIP5 scenarios). These were averaged globally or separately over land and oceans for time series analysis (no correction for drift was performed), and regridded to a common 1-degree grid by linear interpolation for pattern analysis. All

figures of this paper are produced with the Earth System Model Evaluation Tool (ESMValTool) version 2.0 (v2.0) (Righi et al., 2020; Eyring et al., 2020; Lauer et al., 2020), a tool specifically designed to improve and facilitate the complex evaluation and analysis of CMIP models and ensembles.

## 3.1 Global Temperature and Precipitation Projections for Tier 1 and the SSP1-1.9 scenarios


### 3.1.1 Time Series

Figure 1 shows time series of global mean surface air temperature (GSAT) and global precipitation changes (see Fig. A2 in the Appendix for time series of the same variables disaggregated into land-only and ocean-only area averages; also see Tables A3 and A4 for changes under the different scenarios at
mid-century and end-of-the-century). The historical baseline is taken as 1995-2014 (2014 being the last year of CMIP6 historical simulations). The five scenarios presented in these plots consist of the Tier 1 experiments (SSP1-2.6, SSP2-4.5, SSP3-7.0 and SSP5-8.5) and the additional scenario designed to limit warming to 1.5°C above 1850-1900 (a period often used as a proxy for pre-industrial conditions), SSP1-1.9. We smooth each trajectory by an 11-yr running mean to focus on climate-scale variability.
In the plots the thick line traces the ensemble average (see legend and Table A1 for the number of models included in each scenario calculation) and the shaded envelopes represent the 5-95% ranges, which are obtained assuming a normal distribution as 1.64σ, where σ is the inter-model standard deviation of the smoothed trajectories, computed for each year. Only one ensemble member (in the majority of cases r1i1p1f1) is used even when more runs are available for some of the models. By the
end of the century (i.e., as the mean of the period 2081-2100) the range of warming spanned by the multi-model ensemble means under all scenarios is between 0.69 °C and 3.99°C relative to 1995-2014 (0.84°C more when using the 1850-1900 baseline). Considering the multi-model ensemble means as the best estimates of the forced response under each scenario, the range spanned by them can be interpreted as an estimate of scenario uncertainty. When considering the shaded envelopes around the ensemble
mean trajectories, about 0.6°C at the lower end and 1.6°C at the upper end are added to this range. This range can be seen as reflecting the compound effects of model-response uncertainty and some measure of internal variability in the individual model trajectories, but the latter is likely underestimated, given that we are using only one run per model. The use of initial condition ensembles for each of the models would better characterize their respective internal variability (Lehner et al., 2020). Using the 5-95%
confidence intervals as ranges, we find that by the end of the 21[st] century (2081-2100 average, always compared to the 1995-2014 average) global mean temperatures are projected to increase between 2.40°C and 5.57°C for SSP5-8.5, between 1.95°C and 4.38°C under SSP3-7.0, and between 1.27°C and 3.00°C for SSP2-4.5. Global temperatures stabilize or even somewhat decline in the second half of the century in SSP1-1.9 and SSP1-2.6 which span a range from 0.13°C to 1.25°C and 0.40°C to 2.05°C,
respectively, whereas they continue to increase to the end of the century in all other SSPs. The ensemble spread appears to consistently increase with the higher forcing and over time. This suggests that the model response uncertainty increases for stronger responses, an expected result as in higher scenarios and later periods climate sensitivity – which significantly differs among the models -- more strongly influences the model response (Lehner et al. 2020). This result appears robust, given the number of
models included (between 33 and 39 for Tier 1 experiments). Only the number of models contributing to the lowest scenario (SSP1-1.9) is significantly less, i.e., 13 at the time of writing, but the analysis of

ensemble behavior of Sect. 3.2.1 below suggests that for global temperature and precipitation averages ten ensemble members provide a representative sample of the internal climate variability. The same qualitative behavior appears for land-only and ocean-only averages (Fig. A2 and Table A3), with the faster warming over land than ocean reaching on average up to 5.46°C under SSP5-8.5 (compared to the global average reaching 3.99°C) and some models reaching a much larger value under this scenario of 7.57°C. For the lower scenarios, limiting warming in 2100 to 0.69°C and 1.23°C globally translates to an average warming on land of 0.96°C and 1.61°C respectively for SSP1-1.9 and SSP1-2.6 (see Table A3 for all projections and their ranges referenced to the historical baseline).

In order to characterize when pairs of scenarios diverge, we define separation the first occurrence of a positive difference between two time series, one under the higher and one under the lower forcing scenarios, which is then maintained for the remainder of the century. This is similar to Tebaldi and Friedlingstein (2013, TF13 in the following), which used the first occurrence of a significant trend in the year-by-year differences, then justified by the RCPs under consideration, among which only the lowest, RCP2.6, flattened out over the century. In that case, the remainder of the RCPs considered followed an increasing trajectory, with differential rates of increase, therefore justifying the expectation that year-by-year differences would eventually show a significant and persisting trends. Among the new scenarios at least two are expected to follow a flat trajectory, or even a slight peak and decline (SSP1-1.9 and SSP1-2.6) rendering the expectation of a trend in their differences untenable. We therefore adopt a slightly different definition, here, and we also note that this definition would need to be modified if overshoot scenarios -- crossing their reference as they decrease -- were the main focus of this analysis. Also, this is not the only way to define separating scenarios and other studies have applied different, but still fairly similar, definitions, e.g., recently, Marotzke (2019). We use time series of GSAT after applying a 21-year running mean, as we are concerned with differences in climate rather than in individual years, whose temperatures are affected by large variability (this is the part of the definition that takes the place of the consideration of long-term trends in TF13. We also need to choose a threshold by which we deem the difference "positive" and somewhat discernible (this takes the place of asking for a significant trend in TF13). To do so, we use the results in Tebaldi et al., 2015, where the regional sensitivities of temperature and precipitation to changes in global average temperature were quantified. According to that analysis, 0.1°C of difference in 20-yr means of GSAT was the lowest value at which a multi-model ensemble consistently had a positive fraction of the grid-cells experiencing significant warming. In Table A5 we report the precise years when the ensemble means of the smoothed GSAT time series under the various scenario pairs separate according to this definition, and, in parenthesis, when the last of all individual models' pairs of trajectories separate, but of course those precise estimates would change if our choices of the moving window and the threshold had been different. The ensemble average trajectory of GSAT under SSP5-8.5 separates from the lower scenarios' ensemble average trajectories between 2027 and 2034 with the longer time as expected applying to the separation from SSP3-7.0. SSP3-7.0 separates from the two scenarios at the lower end of the range between 2031 and 2037, and ten years later from SSP2-4.5. The ensemble average trajectory of global temperature under SSP2-4.5 separates from those under the two lower scenarios, SSP1-1.9 and SSP1-2.6 by 2034 and 2039 respectively, while the ensemble average GSAT trajectories under the two lower scenarios, SSP1-1.9 and SSP1-2.6, separate from one another in 2042 (in Fig. A3

the differences between ensemble averages for each pair of scenarios appear as red lines). When
considering individual models' trajectories under the different scenarios, and defining the time of
separation when the last of all individual pair of trajectories separates, model structural differences and
a larger effect of internal variability cause a significant delay compared to the ensemble mean
separation. Depending on the pair of scenarios considered, the length of the delay necessary for the last
of the models to show separation varies significantly: as few as 6 years for the full separation of SSP1-
2.6 from SSP5-8.5, as many as 19 years for the full separation of SSP3-7.0 from SSP5-8.5 (Fig. A3,
black lines, and values in parenthesis in Table A5).
Ensemble mean precipitation change by 2081-2100 (as a percentage of the 1995-2014 baseline) is
between 2.0 and 3.0% for the lowest scenarios (SSP1-1.9 and SSP1-2.6), 4.2 and 4.9% for SSP2-4.5
and SSP3-7.0, and 7.3% for SSP5-8.5. As expected, the larger variability of precipitation changes
(relative to temperature changes), both from internal sources and model response uncertainty, is such
that only the highest scenario ensemble mean trajectory separates from the lower ones appreciably
before 2050 while the lowest scenario separates from the rest around mid-century. The ensemble means
of the three scenarios in between overlap until close to 2070. The multi-model spread and internal
variability confound a large fraction of the individual scenarios' trajectories until the end of the century
(Fig. 1, right panel). Both the magnitude of the changes and their variability are larger for precipitation
averages over land than over oceans (Fig. A2; see also Table A4 for a complete list of mid- and late
century changes).

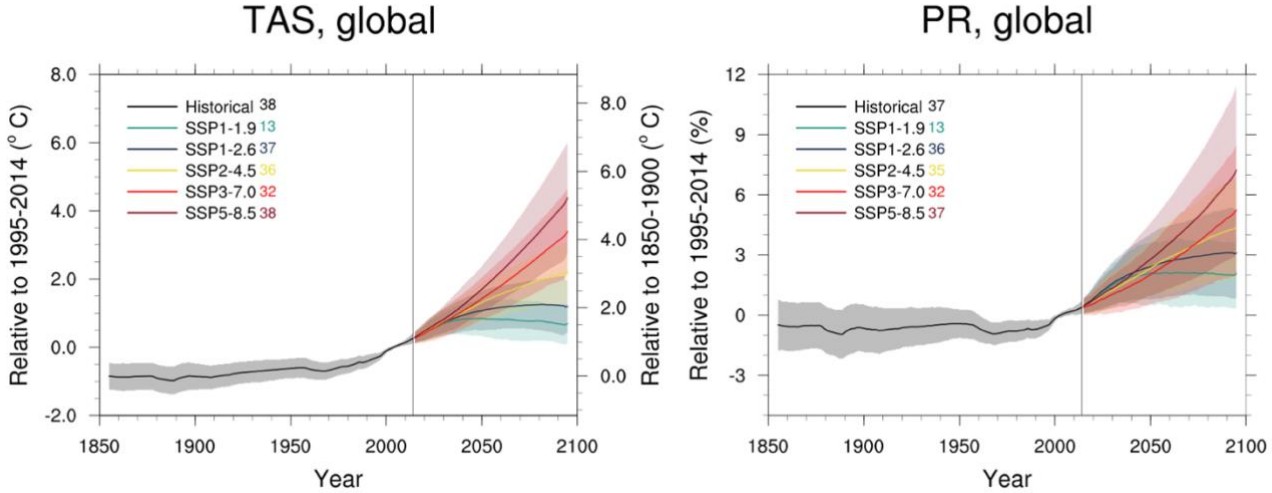

**Figure 1: Left panel: global average temperature time series (11-yr running averages) of changes from current baseline (1995-2014, left axis) and pre-industrial baseline (1850-1900, right axis, obtained by adding a 0.84°C offset) for SSP1-1.9, SSP1-2.6, SSP2-4.5, SSP3-7.0 and SSP5-8.5. Right panel: global average precipitation time series (11-yr running averages) of percent changes from current baseline (1995-2014) for SSP1-1.9, SSP1-2.6, S SP2-4.5, SSP3-7.0 and SSP5-8.5. Thick lines are ensemble means (number of models shown in the legends). The shading represents the +/-1.64σ interval, where σ is the standard deviation of**
**the smoothed trajectories computed year-by-year (thus approximating the 5-95% confidence interval around the mean of a normal distribution). Note that the uncertainty bands are computed for the anomalies with respect to the historical baseline (1995-2014). Thus, the right axis of the global temperature plot, showing anomalies with respect to pre-industrial, applies to the ensemble means, not to the uncertainty bands, which would be narrowest over the period 1850-1900 if we were to calculate uncertainties on**

the basis of the models' output over that period, rather than by simply adding an offset uniformly. See Fig. A2 in the Appendix for
land-only and ocean-only averages and Tables A3 and A4 for the values of changes at mid and late century.

### 3.1.2 Normalized Patterns

In Fig. A4 we show ensemble average patterns of change by the end of the century under the five
scenarios for both variables. In this section we focus our discussion on the general features emerging
from the average *normalized* patterns. Normalized patterns are computed as the end-of-century
(percent) change compared to the historical baseline, divided by the corresponding change in global
mean temperature. This computation is first performed for each individual model/scenario, at each grid
point, after regridding temperature and precipitation output to a common 1°x1° grid. The individual
normalized patterns are then averaged across models and the five scenarios. As we will show, the total
variations among the population of normalized patterns that form this grand average is mainly driven by
inter-model variability, rather than inter-scenario differences. Thus, we choose to synthesize patterns of
change across all scenarios by presenting regional changes per degree of global warming. More in depth
analyses, also exploiting complementary experiments from LUMIP and AerChemMIP, may provide a
more refined view of the inter-scenario differences possibly arising from different regional forcings.
Fig. 2a shows the spatial characteristics of warming, and of wetting and drying. For temperature
changes, the left panel confirms the well-established gradient of warming decreasing from Northern
high latitudes (with the Arctic regions warming at twice the pace of the global average) to the Southern
Hemisphere, and the enhanced warming in the interior of the continents compared to ocean regions
(which consistently warm slower than the global average). This differential is particularly pronounced
in the Northern Hemisphere (and would be muted if the normalized pattern was computed at
equilibrium). The familiar cooling spot in the Northern Atlantic appears as well - the only region with a
negative sign of change. Studies have suggested that the cooling signal is an effect of the slowing of the
Atlantic Meridional Overturning Circulation, which creates a signal of slower northward surface-heat
transport, resulting in an apparent local cooling (Caesar et al., 2018; Keil et al., 2020).
For precipitation, the strongest positive changes are in the equatorial Pacific and the highest latitudes of
both hemispheres, especially the Arctic region. The large changes in subtropical Africa and Asia are due
more to the small precipitation amounts of the climatological averages in these regions (at the
denominator of these percent changes), than to a truly substantial increase in precipitation (see also
below, for variability considerations). A strong drying signal continues to be projected for the
Mediterranean together with central America, the Amazon region, Southern Africa and Western
Australia.
Similar to Tebaldi & Arblaster (2014), we give a measure of robustness of these patterns by computing
the standard deviation at each grid-point across individual model/scenario patterns (Fig. 2b). We further
distinguish the relative contribution of scenario and model variability by computing standard deviations
*after* averaging across models separately for each individual scenario, and across scenarios for each
individual model, respectively. Fig. 2b, top row, highlights in darker colors regions where the standard
deviation is higher and patterns are less robust. For temperature patterns, as has been found in earlier
studies of pattern scaling (starting from Santer et al. (1990) and in more recent work, like Herger et al.
(2015)) the edges of sea ice retreat at both poles are areas where models disagree, and scenarios, in
lesser measure, can be at odds due to their different timing of persistent ice melt. The variability and

therefore uncertainty of the precipitation pattern mirrors the signal of change at low latitudes in the Pacific and over Africa and Asia. The comparison of patterns in the middle and bottom rows of the figure elucidate the role of inter-model variability rather than scenario variability for both temperature

and precipitation normalized changes, with scenario uncertainty only contributing to a small area of sea ice variability in the Arctic for temperature change, and a subregion of the Sahara for precipitation change (where the denominator of the percentage values is small and therefore prone to cause instabilities in the values computed) . Given the radically different sample sizes used to compute the averages from which scenario-driven standard deviations are derived compared to model-driven (more

than 30 for the former, and only 5 for the latter), we can also infer that internal variability is a likely contributor to model-driven standard deviation, while is mostly eliminated before the computation of the scenario-driven standard deviation.

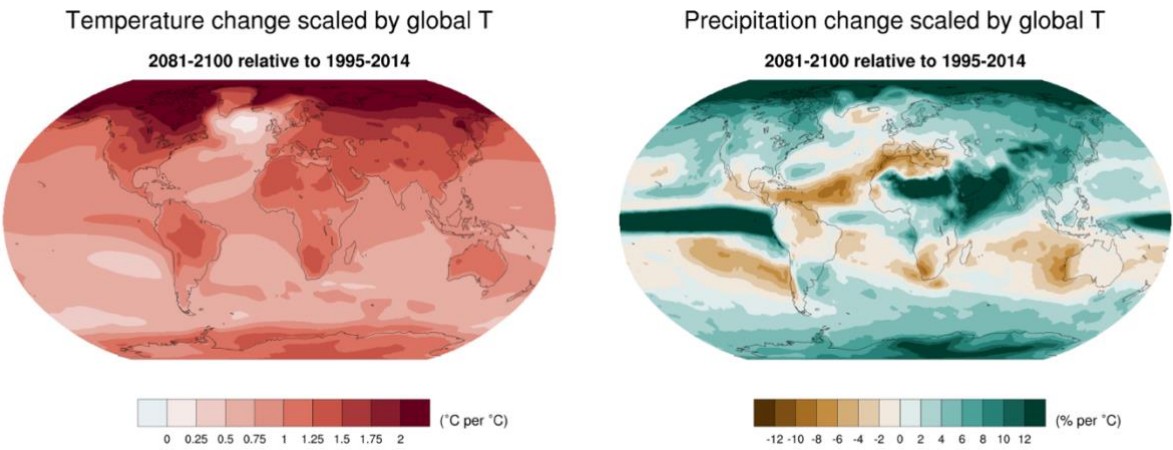


**Figure 2a: Patterns of temperature (left) and percent precipitation change (right) normalized by global average temperature change (averaged across CMIP6 models and all Tier 1 plus SSP1-1.9 scenarios).**

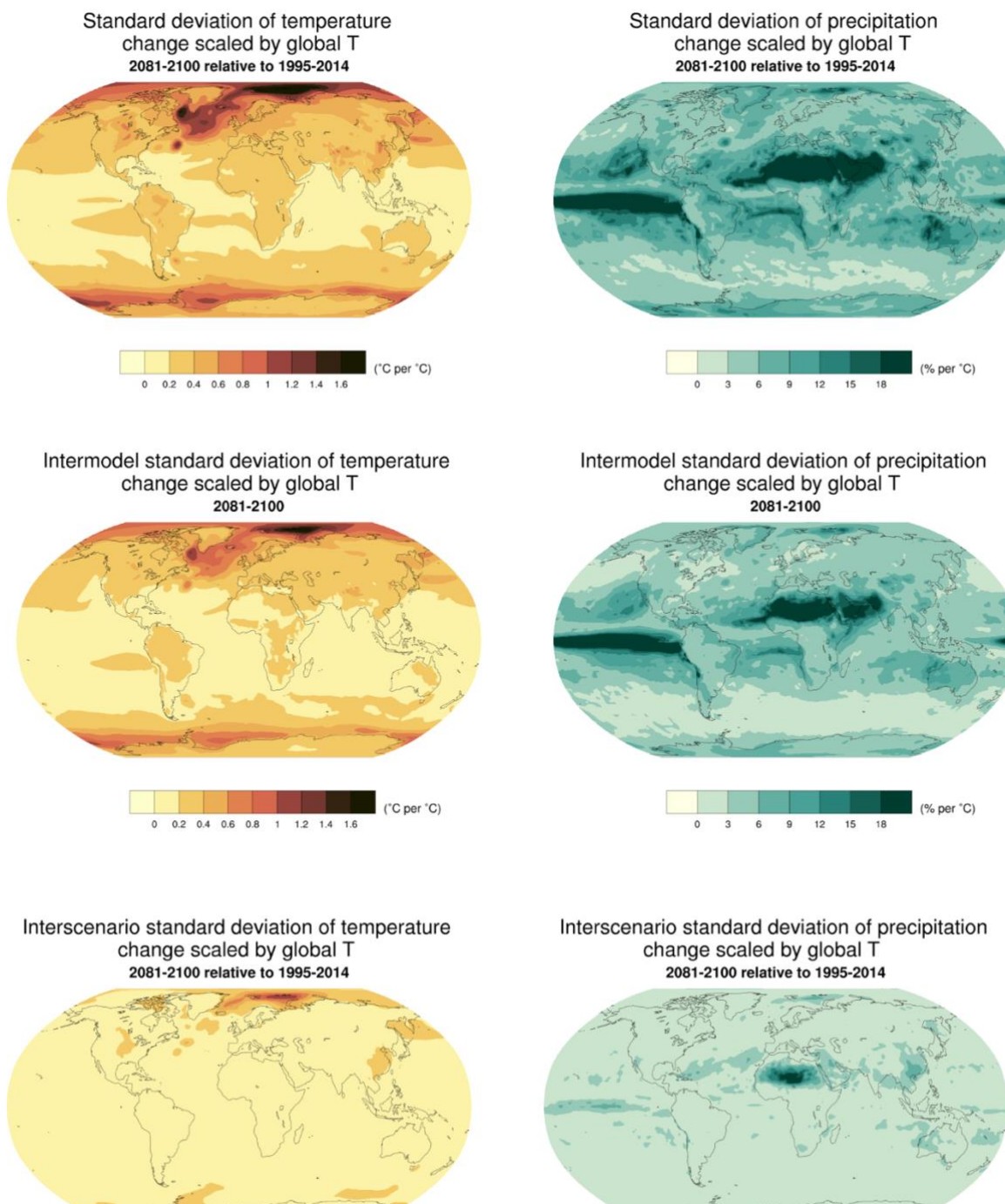

 **Figure 2b: Top row: standard deviation of normalized patterns for individual CMIP6 models and scenarios. The individual patterns are the elements from which the averages shown in Fig. 2 are computed. Center row: Standard deviation of normalized patterns, after averaging across scenarios, highlighting the role of inter-model variability. Bottom row: Standard deviation of normalized patterns after averaging across models, highlighting the role of inter-scenario variability.**


The robustness of these multi-model average patterns and the sources of their variability can be assessed by considering the same type of graphics computed from the four RCPs from the CMIP5 model ensemble.
Figure 3 (top row) and Fig. A5, using the same color scales, are easily compared to Fig. 2a and Fig. 2b
respectively, and confirm the striking consistency of the geographical features of the normalized patterns, the size and spatial features of their variability, together with the components of the latter (i.e., model vs. scenario variability).

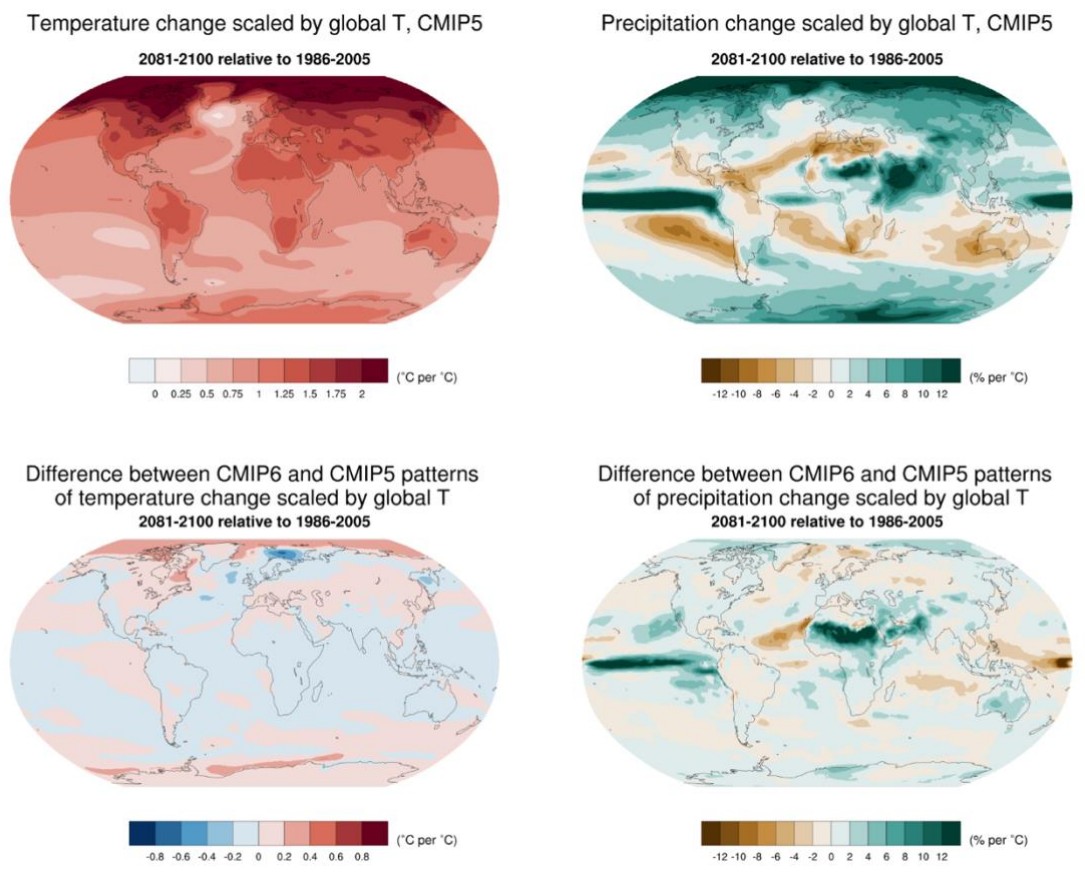

**Figure 3: Patterns of temperature (left) and percent precipitation change (right) normalized by global average temperature change (averaged across models and scenarios) from CMIP5 models and scenarios, for comparison with Fig. 2.**

We deem a rigorous quantification of the differences between patterns beyond the scope of this paper, and focus on a qualitative assessment of the similarities that surface by showing on the bottom row of

Fig. 3 the difference between CMIP6 and CMIP5 normalized patterns, confirming the small magnitude of the discrepancies in TAS over all regions, except for the Arctic, known to be affected by large variations among model, scenarios (with a possible role of the lowest scenario in CMIP6, SSP1-1.9, whose land-sea ratio has likely no equivalent among the CMIP5 scenarios, but further, more rigorous investigation is needed to confirm this) and internal noise (likely playing a minor role given the number of model and scenarios contributing to these averages). Similarly for percent precipitation the regions that stand out where the largest differences are found are the tropics, known to be affected by large variability and uncertainties. In this case the possible role of aerosol forcing (Yip et al., 2011) warrants further investigation, especially as we consider that SSP3-7.0 forcing composition and trajectory are quite different from previous scenarios'. As mentioned, the use of these experiments in conjunction with their variants by LUMIP and AerChemMIP could further attribute some of these scenario-dependent features to differences in regional forcing like land-use or aerosols. Also, a subset of CMIP6 models are running the CMIP5 RCPs, and results from those experiments will allow a clean analysis of variance, partitioning sources between model and scenario generations.

### 3.1.3 Comparison of climate projections from CMIP6 and CMIP5 for three updated scenarios

In the previous section the comparison of normalized patterns was by construction scenario independent. The design of ScenarioMIP, however, deliberately included scenarios aimed at updating CMIP5 RCPs, and three of those are in Tier 1. Updates in the historical point of departure (2015 for CMIP6 rather than 2006 for CMIP5) together with updates in the models forming the ensemble which reflect on the radiative forcing levels simulated by the individual models (Smith et al., 2020) are obvious differences that hamper a straightforward comparison. In addition, the emission composition of the scenarios also changed with the update, and we summarize how after presenting the projection comparison.

We show time series of global temperature for the three updated scenarios and the corresponding results from their CMIP5 counterparts: SSP1-2.6 vs RCP2.6, SSP2-4.5 vs RCP4.5, and SSP5-8.5 vs RCP8.5 from CMIP6 and CMIP5, respectively. We show warming relative to the same historical baseline of 1986-2005 used by CMIP5 (Taylor et al., 2012) and to 1850-1900. We further show how observational constraints applied to the range of trajectories from the new models based on recently published work (Tokarska et al., 2020) result in lower and narrower projections at the end of the century, and have the effect of bringing CMIP6 projections in closer alignment to CMIP5 end-of-the century warming, even when the same type of constraints are applied to the latter.

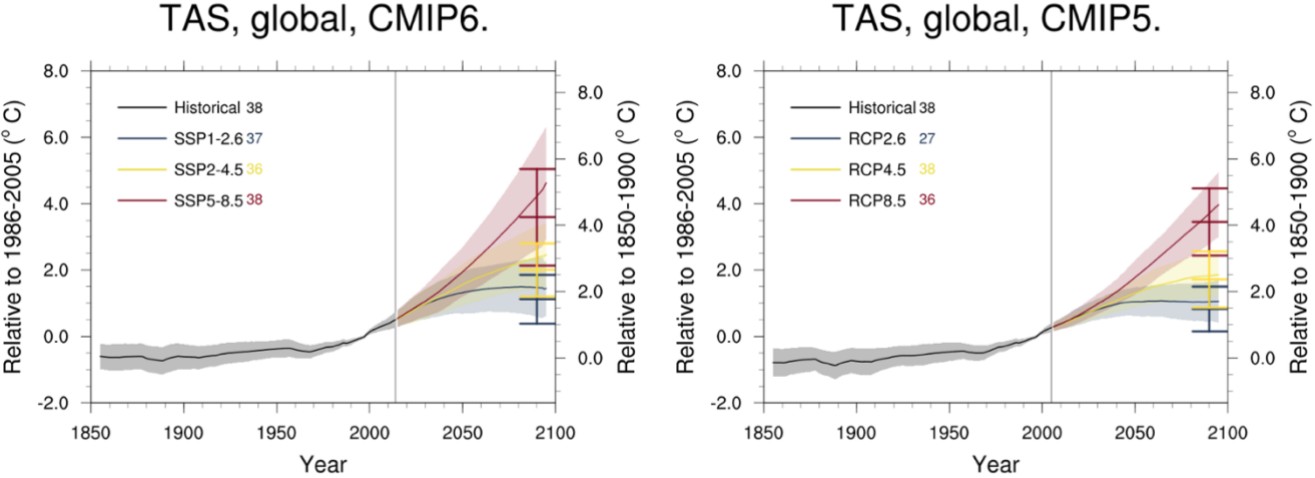

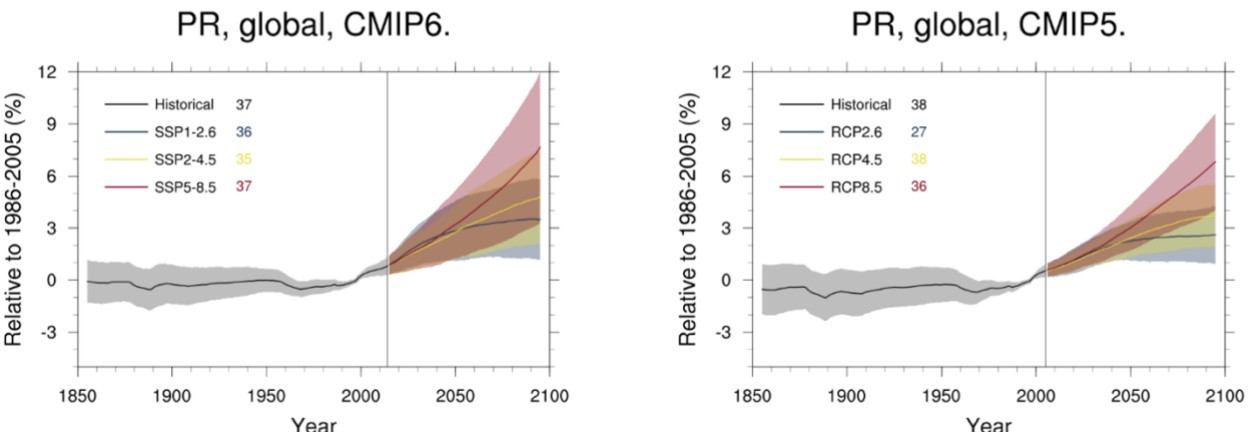

**Figure 4: Comparison of the three SSP-based scenarios updating 3 CMIP5-era RCPs with the corresponding CMIP5 output:**
435 **SSP1-2.6, SSP2-4.5 and SSP5-8.5 on the left can be compared to RCP2.6, RCP4.5 and RCP8.5 on the right for global average temperature change (top row) and global average precipitation change (as a percentage of the baseline values, which are set to 1986-2005 for both ensembles). Indicators along the right axis of the plots of temperature projections show constrained ranges at 2100, obtained by applying the method of Tokarska et al. (2020). Note that, as in Fig. 1, the uncertainty bands in all figures are computed for anomalies with respect to the historical baseline (1986-2005 in this case). Thus the right axis of the global**
440 **temperature plots, showing anomalies with respect to pre-industrial, applies to the ensemble means, not to the uncertainty bands, which would be narrowest over the 1850-1900, were they calculated using the data from simulations over that period, rather than being registered to the new axis only on the basis of the offset. Figure A6 shows a more direct comparison of the CMIP6 and CMIP5 ranges before and after the application of constraints at 2081-2100, and Table A6 lists those ranges (and the unconstrained percent precipitation changes for the same comparisons) at 2041-2060 and 2081-2100.**

445

Figure 4 aligns two pairs of plots showing time series of global temperature and percent precipitation changes under the three updated scenarios and the original RCPs, from the CMIP6 and CMIP5 ensembles respectively: the left-hand panels show three of the trajectories already shown in Fig. 1 (left

panels of both rows) but as anomalies/percent changes from the period 1986-2005, i.e., the last 20 years
of the CMIP5 historical period (Taylor et al., 2012). The right-hand side panels show CMIP5 results for
the three corresponding RCPs (see Table A2 for a list of the models used), also using the 1986-2005
baseline. The right axis on the temperature plots allows an assessment of changes compared to the
1850-1900 baseline. Table A6 lists mid- and late century changes for all model ensembles under the
different scenarios. The new *unconstrained* results reach on average warmer levels, and have a larger
inter-model spread, especially when comparing SSP5-8.5 to RCP8.5. There is 0.46°C (for the scenarios
reaching 2.6Wm$^{-2}$), 0.49°C (for the 4.5Wm$^{-2}$ scenarios) and 0.67°C (for the 8.5Wm$^{-2}$ scenarios) more
mean warming, while the upper end of the shading for SSP5-8.5 reaches 1.5°C higher than the CMIP5
results (Table A6). The larger warming resulting from the CMIP6 experiments is a combination of
different forcings and the presence among the new ensemble of models with higher climate sensitivities
than the members of the previous generations. The higher climate sensitivities in CMIP6 compared to
CMIP5 (Meehl et al., 2020; Zelinka et al., 2020) become more critical for higher forcings, when the
model response is more highly correlated to its climate sensitivity explaining the differential in the
higher warming across the range of new scenarios, with the largest difference evident for SSP5-8.5.
Several recent studies (Brunner et al., 2020, Liang et al., 2020, Nijsse et al., 2020; Ribes et al., 2020 and
Tokarska et al., 2020) constrain the ensemble projections according to the evaluation of the ensemble
historical behavior. All studies find a strong correlation between the simulated warming trends over the
observed historical period and the warming in SSP scenarios, which suggested constraining future
warming using observed warming trends estimated from several observational products, and all come to
similar results. Here and in Table A6 we show how the 2081-2100 means for both CMIP5 and CMIP6
are changed as a result of applying constraints as in Tokarska et al. (2020). Also in Fig. A6 we show the
same results, but focusing specifically on these 20-yr means, before and after the application of the
constraints. The resulting observationally-constrained ranges bring CMIP6 projections closer to both the
the raw CMIP5 ranges and their constrained counterparts in both mean and spread (especially the upper
bound). In other words, models that project the most warming by the end of the century tend to do the
least well in reproducing historical warming trends for both ensembles, but the effect is much more
pronounced for CMIP6 than CMIP5 models (see also Fig. A6). After constraints are applied, the
difference in the mean changes by 2081-2100 is 0.29°C for the two lower scenarios, and 0.15°C under
SSP5-8.5/RCP8.5. The difference in the upper range under the latter scenario is reduced to 0.59°C
Global precipitation projections follow temperature projections (O'Gorman et al., 2012), and therefore
we see (unconstrained) CMIP6 trajectories reaching higher percent changes than CMIP5 of just below
1%. Consistent with the relatively larger means, the spread of trajectories for individual scenarios,
which combines internal variability with model uncertainty, is larger for the new models and scenarios.
As mentioned, part of the differences described are due to forcing differences between the
corresponding scenarios in CMIP5 and CMIP6. These are by design small in terms of aggregate
radiative forcing, when radiative forcing is defined as IPCC-AR5-consistent total global stratospheric
adjusted radiative forcing (AR5-SARF). By this measure of forcing, scenarios differ by less than 6% in
2100 for the RCP2.6-SSP1-2.6 pair, 5% for the RCP4.5-SSP2-4.5 pair and around 0.3% at 8.9 Wm$^{-2}$ for
the RCP8.5-SSP5-8.5 pair. Differences over the full pathway from 2015 to 2100 are below 15%, 5%
and 4%, respectively. However, the literature in recent years has moved away from the AR5-SARF
definition (in particular, Etminan et al., 2016 – see also implementation in Meinshausen et al., 2020),

towards the use of effective radiative forcing (ERF), which differs from AR5-SARF in that it includes any non-temperature mediated feedbacks (see e.g., Smith et al., 2020).

Given that CMIP5 and CMIP6 concentration pathways differ with respect to their composition across gases and other radiatively active species (Lurton et al., 2020, Fig.1), whose respective ERFs can be very different despite a similar AR5-SARF, the similarity between RCP and SSP scenarios in terms of forcing deteriorates when moving away from an AR5-SARF definition. For example, in SSP5-8.5 the AR5-SARF contribution of $CH_4$ is by 2100 about 0.5 $Wm^{-2}$ lower than in the CMIP5 RCP8.5 pathway. This is offset by the difference in $CO_2$ AR5-SARF, where SSP5-8.5 is around 0.5$Wm^{-2}$ higher. In contrast, these compensating effects do not hold any longer when using ERF. In fact, because ERF is higher than AR5-SARF for $CO_2$ and even more so for $CH_4$, the 2100 radiative forcing level after which both the RCP and SSP pathway are named are not met precisely anymore when measured by ERF. Another pronounced difference between the CMIP5 RCPs and the new generation of SSP-RCP scenarios is that the latter span a wider range of aerosol emissions and corresponding forcings. The main reason for this difference is a wider consideration of the possible development of air pollution policies, ranging from major failure to address air pollution in the SSP3-7.0 pathway to very ambitious reductions of air pollution in the SSP1-2.6, SSP1-1.9 as well as SSP5-8.5 pathways (Rao et al., 2017). All the CMIP5 RCPs followed by comparison a more "middle of the road" pollution policy path. Last, the effective radiative forcing levels reached by both sets of pathways can be different - depending on each climate model processes - from their nominal AR5-SARF values labeling the pathway, usually obtained by running the emission pathways through simple models, like using MAGICC in its AR5-consistent setup (Riahi et al., 2017). A recent study with the EC-Earth model finds that about half of the difference in warming by the end of the century when comparing CMIP5 RCPs and their updated CMIP6 counterparts is due to difference in effective radiative forcings at 2100 of up to 1 $Wm^{-2}$ (Wyser et al., 2020). Fig. A7, adapted from Meinshausen et al., (2020) shows a break-down of the comparison into the three main forcing agents among greenhouse gases, $CO_2$, $CH_4$ and $N_2O$, from which the significant differences in the composition can be assessed. Next to the AR5-consistent SARF time series, we also show effective radiative forcing ranges under the SSPs for the end of the 21st century for comparison using a newer version of MAGICC, MAGICC7.3.

Here we note that in an effort to make the comparison more direct, CMIP5 RCP forcings are available to be run with CMIP6 models, and several modeling centers have started -- at the time of writing -- these experiments, which have been added to the Tier 2 design of ScenarioMIP since the description in O'Neill et al. (2016). If enough models contribute these results, a cleaner comparison of the effects of the updated forcing pathways, controlling for the updated models' effect, will be possible. Preliminary results with the Canadian model, CanESM2, confirm the significant role of higher radiative forcings found with EC-Earth.

### 3.1.4 Scenarios and Warming Levels

The ever-increasing attention to warming levels as policy targets, also due to the recognition that strong relations are found between them and a large set of impacts, motivates us to identify the time windows at which the new scenarios' global temperature trajectories reach 1.5, 2.0, 3.0, 4.0 and 5.0°C since 1850-1900. Table 1 shows the timing of first crossing of the thresholds by the ensemble average and the 5-95% uncertainty range around that date. This is derived by computing the 5-95% range for the

ensemble of trajectories of GSAT, and identifying the dates at which the upper and lower bounds of the range cross the threshold. The range is computed by assuming a Normal distribution for the ensemble,
as the inter-model standard deviation multiplied by 1.64. Considering this range rather than the minimum and maximum bounds of the ensembles makes the estimates of the 5-95% range more robust, especially for the lowest scenario, SSP1-1.9 for which we only rely on 13 models. The analysis is conducted after smoothing each of the individual models' time series by an 11-year running average, to smooth interannual variability. The width of the intervals would change if constraints based on the
observed warming trends were applied to the ensemble along the whole century (as shown in Fig. 4 for the end of the century) but here the unconstrained ensemble is used. The anomalies from 1850-1900 are computed as described in Sect. 3.1.1, by computing anomalies with respect to the historical baseline (1995-2014) and then adding the offset value of 0.84°C to minimize the effect of biases in the warming during the historical period of the different models. Note however that remaining differences between
models and observations in the warming trends over the period 2014 to present, and the effects of differences between observed and projected forcings may still introduce biases in the crossing level estimates, likely biasing them low.
We first synthesize results from the experiments from Tier 1, for which we extract a common subset of 31 models in order to make the threshold crossing estimates comparable across scenarios (for
completeness we document in Table A7 the behavior of all models available, which does not change qualitatively the results that we are about to describe).
The lowest warming level of 1.5°C from pre-industrial is reached on average between 2026 and 2028 across SSP1-2.6, SSP2-4.5, SSP3-7.0 and SSP5-8.5 with largely overlapping confidence intervals that start from 2020 as the shortest waiting time and extend until 2046 at the latest under SSP2-4.5. Note
however that the lower bound of the ensemble trajectories (determining the upper bound of the projected years by which the level is reached) under SSP1-2.6 does not warm to 1.5°C for the whole century (the NA as the upper bound of the time period signifies "not reached"). The next level of 2.0°C is reached as soon as 13 years later by the ensemble average under SSP5-8.5, and as late as 32 years later under SSP1-2.6, a striking reminder of how different the pace of warming is in these scenarios.
The confidence intervals have similar lower bounds between 2030 and 2032 and extend to 2077 for SSP2-4.5, while they are significantly shorter for the higher scenarios (2064 and 2054 for SSP3-7.0 and SSP5-8.5 respectively). The confidence intervals for SSP1-2.6 do not reach any of the higher warming levels, while by 2059 the ensemble average under SSP5-8.5 has already warmed by 3°C. SSP3-7.0 takes 9 more years, while it takes until 2092 for the ensemble average under SSP2-4.5 to reach 3°C. Under
this scenario it is worth noting that only 21 out of 37 models reach that level. Only the ensemble means of the two higher scenarios reach 4°C, as early as 2077 for SSP5-8.5, and 14 years later for SSP3-7.0. The highest warming level considered of 5°C is only reached by the upper range of SSP3-7.0 (only 4 models out of 33) while more than half the models running SSP5-8.5 (21 out of 39) reach that warming level in the last decade of the century (2094) as an ensemble average, and as early as 2074 when the
warmer end of the ensemble range is considered.
Only 13 models are available at the time of writing under the lowest scenario specifically designed to meet the Paris Agreement target of 1.5°C warming by the end of the century. Of those, two remain below that target for the entire century, while others have a small overshoot of the target which was expected by design. The ensemble mean reaches 1.5°C already by 2029. The lower bound never crosses

that level, while the upper bound is already at 1.5°C currently, i.e., by 2021 (as a reminder, CMIP6 future simulations start at 2015). In Table A8 in the Appendix, a comparison of the CMIP5/CMIP6 three corresponding scenarios (SSP1-2.6, SSP2-4.5 and SSP5-8.5 compared to RCP2.6, RCP4.5 and RCP8.5) for a slightly larger ensemble of 36 CMIP6 models for which the three scenarios are available, and a CMIP5 ensemble of 29 models, shows dates compatible with the warmer characteristics of the
CMIP6 models/scenarios. On average, the same target is reached from 3 to 9 years earlier by the CMIP6 ensemble means compared to the CMIP5 ensemble means. A more in depth analysis than is in our scope is necessary to fully characterize the causes of this acceleration. Here we note that the behavior of the CMIP6 ensemble means reflect the use of unconstrained projections, with equal weight given to high climate sensitivity models, which are often also those less adherent to historical trends and that
may show a faster historical warming in the last decade or so than observed. In addition, as we discussed in the previous section, even scenarios having the same AR5-SARF label see different forcings at play. The result is to make the pace of warming faster, and, in several cases, a target that was not reached by the CMIP5 models under a given scenario is instead reached by the corresponding CMIP6 ensemble/scenario. E.g., 2.0°C under SSP1-2.6 is reached in mean in 2056, while it was reached
only by the upper bound (by 2040) under RCP2.6; at the opposite end, 5.0°C was reached only by the upper bound (in 2083) under RCP8.5, while it is reached by the ensemble mean in 2093 under SSP5-8.5.

**Table 1: Times (best estimate and range - in square brackets - based on the 5-95% range of the ensemble after smoothing the trajectories by eleven-year running means) at which various warming levels (defined as relative to 1850-1900) are reached according to simulations following, from left to right, SSP1-1.9, SSP1-2.6, SSP2-4.5, SSP3-7.0 and SSP5-8.5. Crossing of these levels are defined by using anomalies wrt 1995-2014 for the model ensembles and adding the offset of 0.84°C to derive warming from pre-industrial.**
**We use a common subset of 31 models for the Tier 1 scenarios, and all available models (13) for SSP1-1.9, while Table A7 in the**
**appendix shows the result of using all available models under each scenario. The number of models available under each scenario and the number of models reaching a given warming level are shown in parentheses. However, the estimates are based on the ensemble means and ranges computed from all the models considered (13 or 31 in this case), not just from the models that reach a given level. An estimate marked as NA is to be interpreted as "not reaching that warming level by 2100". In cases where the ensemble average remains below the warming level for the whole century, it is possible for the central estimate to be NA, while the**
**earlier time of the confidence interval is not, since it is determined by the warmer end of the ensemble range.**

|  | SSP1-1.9 | SSP1-2.6 | SSP2-4.5 | SSP3-7.0 | SSP5-8.5 |
|---|---|---|---|---|---|
| **1.5°C** | 2029 [2021,NA] (11/13) | 2028 [2020,NA] (30/31) | 2028 [2020,2047] (31/31) | 2028 [2020,2045] (31/31) | 2026 [2020,2040] (31/31) |
| **2.0°C** | NA [2036,NA] (2/13) | 2064 [2032,NA] (17/31) | 2046 [2032,2082] (31/31) | 2043 [2031,2064] (31/31) | 2039 [2030,2055] (31/31) |
| **3.0°C** | NA [NA,NA] (0/13) | NA [NA,NA] (0/31) | 2094 [2058,NA] (16/31) | 2069 [2052,NA] (31/31) | 2060 [2048,2083] (31/31) |
| **4.0°C** | NA | NA | NA | 2091 | 2078 |

| | | | | | |
|---|---|---|---|---|---|
| | [NA,NA] (0/13) | [NA,NA] (0/31) | [NA,NA] (1/31) | [2071,NA] (17/31) | [2062,NA] (27/31) |
| **5.0°C** | NA [NA,NA] (0/13) | NA [NA,NA] (0/31) | NA [NA,NA] (0/31) | NA [2088,NA] (3/31) | 2094 [2075,NA] (15/31) |


## 3.2 Climate projections from ScenarioMIP Tier 2 simulations

### 3.2.1 SSP3-7.0 Initial Condition Ensembles

Five models (CanESM5, IPSL-CM6A-LR, MPI-ESM1-2-HR, MPI-ESM1-2-LR and UKESM1)
contributed at least ten initial condition (IC) ensemble members under SSP3-7.0. We focus here on the behavior of the ensemble spread over the 21st century, as measured by the values of the inter-realization standard deviations. In the following the phrase "ensemble spread" is used, which has to be interpreted as the value of such standard deviation. Fig. 5 shows the time evolution (over 1980-2100) of the ensemble spreads for global temperature and precipitation computed on an annual basis (top row) and
after smoothing the individual time series by an 11-yr running mean (bottom row). One of the models, CanESM5, provides 50 ensemble members that we use to randomly select subsets of 10 members and form a background "distribution" of the timeseries of ensemble spreads, shown in grey in Fig. 5. This is not meant to provide a quantitative assessment but rather a qualitative representation of the variability of "10-member ensembles", which is what most models provide. When we compute trends for the time
series of the temperature ensemble spread all show a negative slope, indicating that the ensemble spread has a tendency to narrow over time. In the case of the spread computed among annual values, only two of the models pass a significance test at the 5% level, while for decadal averages all models show significantly decreasing spreads (significantly negative trends). Trends of the ensemble spreads for precipitation are non-significant for all models when the spread is computed from annual values, while
all are significantly negative, indicating a decrease in the spread, when that is computed from decadal means. This result appears robust for this small set of models, but confirmation with a larger number of models providing sizeable initial condition ensembles will be important. Decreases in GSAT variability have however been found in earlier studies (Huntingford et al., 2013; Brown et al., 2017) and attributed to reduced equator to pole gradients, and reduced albedo variability due to the disappearance of snow
and sea-ice. A deeper investigation of the sources of changes in variability for both variables (which could also tackle how much of the changes in precipitation variability is directly connected to that of GSAT, and what other sources may be at play) is beyond our scope but will be facilitated by the availability of these CMIP6 IC ensembles in addition to the already well studied CMIP5-era large IC ensembles (Deser et al., 2020).
After detrending the values, we compare the distribution of the ensemble spreads for an individual model to that of other models in order to assess if models produce ensembles with spreads that are significantly different. We use a Kolmogorov-Smirnov test (at 5% level) which measures differences in distribution. For several pairs of models, ensemble spreads based on annual values turn out to be indistinguishable: for temperature, CanESM5 ensemble spread is not significantly different from those
of the MPI-ESM model at Low Resolution and those of the UKESM1 model. The latter in turn has an

ensemble spread that is not different from that of the IPSL-CM model. For precipitation, CanESM5 and IPSL-CM produce comparable spreads, as do the two MPI-ESM models, and the MPI-ESM at Low Resolution compared to UKESM1. When we test the spreads of decadal means, all models appear significantly different from one another.  Last, we can exploit the CanESM5 large ensemble in order to assess the number of ensemble members necessary to estimate the forced response of globally averaged TAS and PR, assuming that the mean response obtained by averaging the full ensemble of 50 member is representative of the true forced response. It is found that, for temperature, ten ensemble members produce an ensemble mean trajectory indistinguishable from the one obtained averaging 50 members. For precipitation, only year-to-year variability is not completely smoothed out by averaging ten rather than 50 ensemble members, but filtering by an 11-year running mean effectively cancels out annual "wiggles".

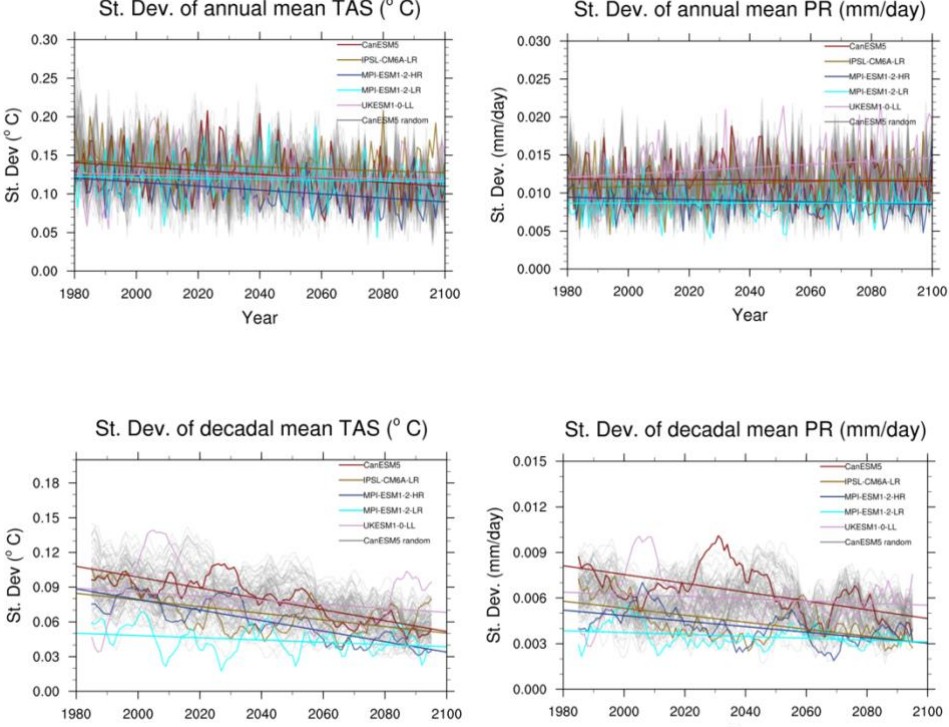

**Figure 5: Time series of ensemble spreads (i.e., inter-member standard deviations) computed at each year among annual (top row) or decadal (bottom row) mean values of TAS (left) and PR (right). The grey lines are obtained by resampling subset of ten members from the CanESM5 model ensemble that provides 50 members. They are meant to provide a qualitative indication of the variability "hidden" in the 10 member ensembles provided by the majority of the models. The color lines show the time series of standard deviations computed from 10 members of 5 models running SSP3-7.0: CanESM5 (first ten members, red), IPSL-CM6A-LR (yellow), MPI-ESM1-2-HR (blue), MPI-ESM1-2-LR (cyan) and UKESM1 (light purple). Straight lines show  least square fits of the linear trends.**

### 3.2.2 Effects of mitigation policies comparing SSP5-8.5 with SSP5-3.4OS, and SSP4-6.0 with SSP4-3.4

The ScenarioMIP design includes two pairs of scenarios, each of which is derived from the same SSP
and integrated assessment model and consists of one baseline scenario without mitigation and one
scenario assuming mitigation policies that reduce radiative forcing. They can therefore be used to
cleanly attribute differences in climate outcomes to mitigation efforts. The two sets of scenarios are
SSP4-6.0 and SSP4-3.4 (produced with the GCAM model, Calvin et al., 2017), and SSP5-8.5 and
SSP5-3.4OS (produced with the ReMIND-MagPIE model, Kriegler et al., 2017). Fig. 6 and Fig. 7 show
time series of global temperature and percent precipitation anomalies with respect to the baseline period
of 1995-2014 for the two pairs, and the patterns of differences in temperature and percent precipitation
changes by the end of the century, which we can characterize as the benefits of mitigation within the
two SSP worlds. For reference, the pattern of change for the lower scenario in the pair is also shown.
Figure 6 shows these outcomes for the pair of scenarios developed under SSP5. One of them is the
unmitigated pathway already featured in the previous sections, SSP5-8.5, assuming high reliance on
fossil fuels to support economic development, and reaching 8.5Wm$^{-2}$ by the end of the century. The
other scenario, SSP5-3.4OS, follows the same path of emissions until 2040, when it enforces a steep
decline in greenhouse gas emissions, which become negative after 2070 and therefore create an
overshoot in concentrations, radiative forcing and global average temperature, to end up at 3.4Wm$^{-2}$ at
2100. Note that the end-point of this scenario, according to these global measures, coincides with the
end-point of SSP4-3.4, the lower scenario of the other pair considered in this section, which is however
reached along a traditional non-exceed pathway.
Figure 7 shows results for the other pair, developed under SSP4, which by the end of the century
reached 6.0Wm$^{-2}$ (without mitigation) and 3.4Wm$^{-2}$ (with mitigation) respectively. Their greenhouse
gas emissions start diverging immediately, by 2020, with those of the lower scenario already
decreasing by that time, while those of the baseline scenario continue to increase for two more decades,
plateauing and then decreasing only after 2060. Both scenarios have a non-decreasing shape in radiative
forcing and temperature.
At global scales, Fig. 6 and Fig. A8 (left panel) show that the forced temperature signals (identified by
the ensemble averages, i.e., the red lines in the time series separation plots) for the SSP5-driven
scenario pair respond within a decade of the divergence in the emission pathways, i.e., they separate by
2050 (just a couple of years later if we consider the last of the individual models) when we apply the
same definition of separation used in Sect. 3.1.1. Global percent precipitation changes show the
expected delay in the emergence of the mitigation signal, with ensemble average time series separating
only after 2060 and the overlap of a large fraction of individual ensemble members under the two
scenarios persisting until the end of the century. The corresponding time series in Fig. 7 (and the middle
panel of Fig. A8) shows that separation of temperature trajectories takes place even earlier for this pair
of scenarios, by 2040 (2045 for the last of the individual models), consistently with the earlier start of
the mitigation. A large majority of the precipitation trajectories still overlap at the end of the century.
The differential patterns of temperature and precipitation change have strikingly similar spatial features
when comparing Fig. 6 and Fig. 7, only modulated by the strength of the changes, proportional to the
gap in radiative forcings. Temperature changes benefit from mitigation over the whole globe, but more
significantly and increasingly so the higher the latitude in the Northern Hemisphere. All land regions
see a benefit of mitigation (in terms of the forced signal, again represented by the difference in
ensemble mean changes) of at least 2°C to 3°C in annual average temperatures at the end of the century,

larger in most of the NH land regions and reaching 8°C in the Arctic for the SSP5-3.4OS/SSP5-8.5 scenario pair. For precipitation changes, the larger differences translate in a more than doubled intensity (note that the colors are the same or stronger in the difference plot than in the scenario change plot) in both directions of change over the high latitudes (wetting) and the subtropics (drying). It is worth

pointing out that patterns of change under the individual scenarios and patterns of differences between scenarios are similar, a further indication of the stable nature of the patterns of future change across different forcing scenarios.

Last, we use Fig. 6 and Fig. 7, together with the third panel of Fig. A8 for an additional comparison, as the presence of two scenarios ending at the same level of radiative forcing (AR5-SARF), SSP4-3.4 and

SSP5-3.4OS, allows us to compare the effects of the overshoot, after performing the same differencing for the 6 models that ran both of these scenarios. A comparison of the patterns of change under the two scenarios shows no apparent differences in the intensity of the changes for both temperature and precipitation, consistent with the global time series reaching a similar warming and precipitation change level at 2100. The model by model differences of these two scenarios (right panel in Fig. A8) for

temperature show that the effects of the overshoot trajectory translate in warmer global temperatures starting from 2032 and all the way to 2080 in the ensemble mean, and from 2038 to 2087 when considering the least differentiated of the individual models' pairs. The overshoot causes 0.4°C of additional warming in the middle of the 2030-2080 period (2055), with a cumulative measure of differential warming over the period of about 14 degree-years. This simple analysis suggests that

average temperature and precipitation changes do not show significant memory and recover quickly after an overshoot of this magnitude.

The small number of models supporting these conclusions leaves the possibility that some of these numbers could change, when larger multi-model ensembles will become available.


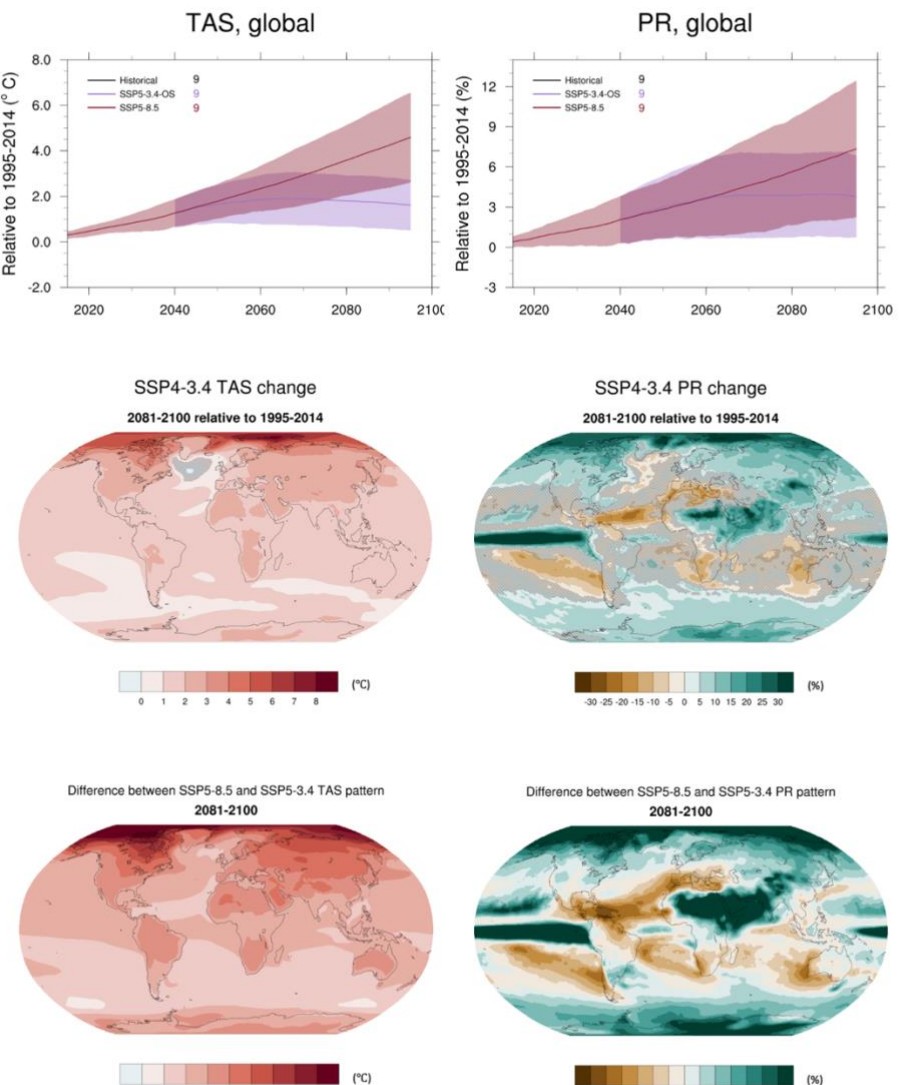


**Figure 6: Time series and patterns comparing SSP5-8.5 to SSP5-3.4OS. First row: Global average time series of temperature and percent precipitation change with respect to the 1995-2014 baseline (11-yr running means). Second row: Patterns of change for the same quantities, under the lower scenario, SSP5-3.4OS (stitched areas are not significant, i.e., the magnitude of the change does not exceed the models' standard deviation). Third row: Differences between the patterns of change under the higher (SSP5-8.5)**
**and lower scenario.**

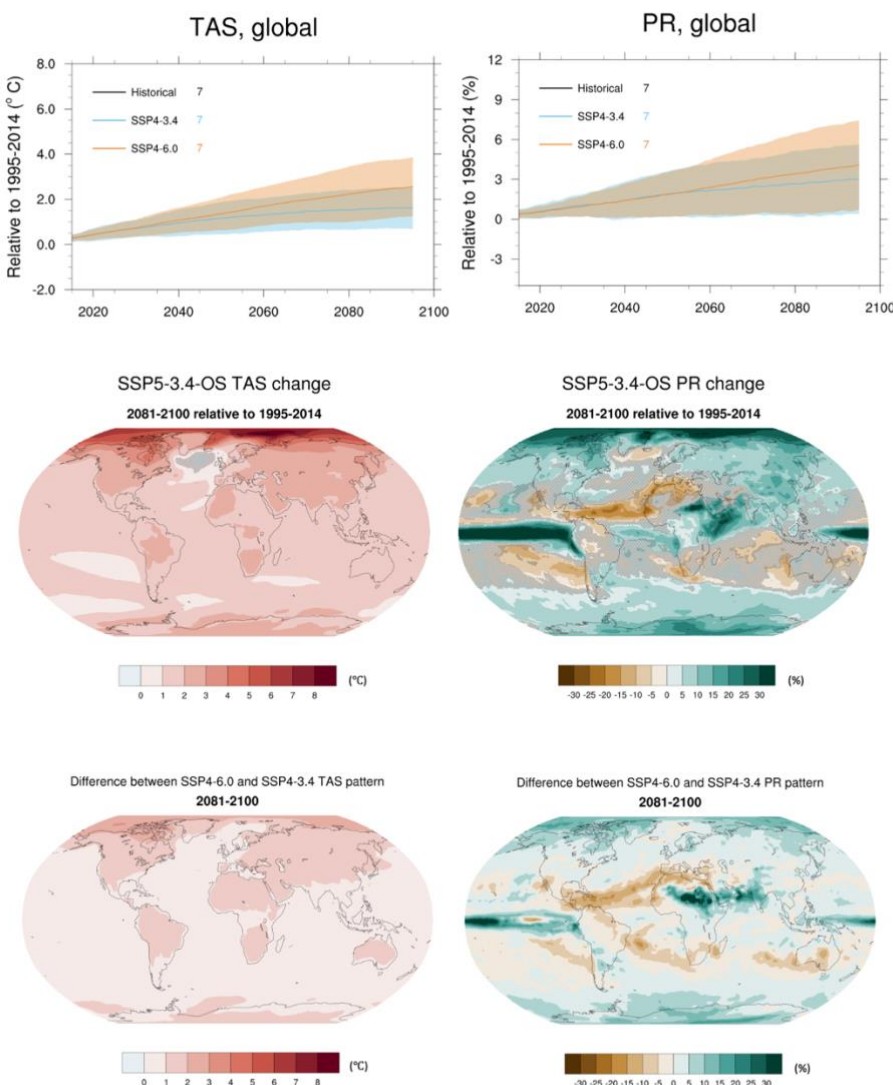

**Figure 7: Like Fig. 6, but for SSP4-6.0 and SSP4-3.4.**

## 4. Summary and Discussion

This paper provides an overview of ScenarioMIP results for surface temperature and precipitation projections under both Tier 1 and Tier 2 experiments, in addition to a comparison to CMIP5 outcomes for a subset of experiments that updated three of the RCPs.

The number of models contributing results for the simulations of 21st century scenarios ranges from
almost 40 for experiments in Tier 1 to only 7 for some of the experiments in Tier 2. At the time of
writing the availability of the long-term simulations results is too scarce to provide a robust multi-model
ensemble perspective and we have not included those results.
Ensemble mean trajectories of global temperature under the Tier 1 and the 1.5°C scenarios (SSP1-1.9,
SSP1-2.6, SSP2-4.5, SSP3-7.0 and SSP5-8.5) span values between 0.7°C and 4.0°C above the historical
baseline (1995-2014) (1.5°C-4.8°C above 1850-1900 average), but individual models reach
significantly larger warming levels under SSP5-8.5, above 5.5°C (6.4°C from 1850-1900). A
comparison with the three CMIP5 RCPs (RCP2.6, RCP4.5 and RCP8.5) which reach the same nominal
level of radiative forcing in 2100 (in terms of AR5-SARF) shows a wider range covered in the newest
simulations, especially with respect to the upper end. Studies (Tokarska et al., 2020; Nijsse et al., 2020;
Brunner et al., 2020; van Vuuren and Carter, 2014) have confirmed that this is attributable to an
interplay of both higher radiative forcings by 2100 in the scenarios (when measured by the currently
preferred metric, ERF) and higher climate sensitivities in a subset of the CMIP6 models, together with
differences in background volcanic aerosols and greenhouse gases that make a straightforward
comparison not possible (Fyfe et al., 2020; Lurton et al., 2020; Meehl et al., 2020; Meinshausen et al.,
2020; Michou et al., 2020; Nicholls et al., 2020, Séférian et al., 2020; Smith et al., 2020, Wyser et al.,
2020). We have shown that when applying constraints based on historical warming rates that weigh
models differently on the basis of their performance (Tokarska et al., 2020), ensemble means and ranges
of the CMIP6 experiments are brought closer to the corresponding means and ranges from CMIP5
model results, as many of the models with higher climate sensitivities also tend to perform less well
over the historical period in terms of regional and aggregate warming trends (Brunner et al., 2020). This
remains true even when the same constraints are applied to the CMIP5 ensembles, as they do not have
as large an effect on the resulting trajectories (Fig. 4 and Fig. A6). A recent assessment performs a
thorough attempt at constraining the distribution of climate sensitivity based on multiple lines of
evidence, independently of climate models characteristics (Sherwood et al., 2020). If the resulting
distribution of ECS were to be used to downweigh or cull models whose ECS is deemed an outlier, we
would likely see changes in the CMIP6 ensemble projections in the same direction as those obtained by
historical warming constraints, but formal studies applying this alternative type of constraints have not
yet been published.  The lack of a one-to-one correspondence between ECS and Transient Climate
Response (Sanderson, 2020), the latter more directly responsible for transient warming, further urges
caution with this inference. According to the Tier 1 scenarios and SSP1-1.9 the 1.5°C target (above
1850-1900) is reached by the model ensemble average in the second half of the current decade (between
2026 and 2029 depending on the scenario). The scenario decides if the 2.0°C threshold is reached after
only 13 more years (SSP5-8.5) or after more than 35 (SSP1-2.6) whereas it is never reached under
SSP1-1.9. Only under SSP3-7.0 and SSP5-8.5 does a majority of models reach 4°C, while 5°C is
reached by half of the ensemble members only under SSP5-8.5: models produce 4.0°C of warming, on
average, under the two higher scenarios in 2078 (SSP5-8.5) and 2091 (SSP3-7.0), while by 2094 5.0°C
is reached by the ensemble average under SSP5-8.5. Global precipitation change follows the pace and
magnitude of warming (O'Gorman et al., 2012; Lambert et al., 2008) and spans a higher range of
ensemble mean projections (by slightly less than 1%) than CMIP5 and a wider range of variability
around them. Time series computed separately for land and ocean regions, and global patterns of change

- calculated as function of global warming - confirm well established behaviors: warming is stronger over land than over oceans; the North to South warming gradient over the globe persists, with strong polar amplification signals resulting in projected warming at twice the pace of the global average in the Arctic region. The regional cooling effect of North Atlantic upwelling emerges clearly. Precipitation
change appears appears stronger on average over land than over the globally averaged oceans, with the (by now familiar) multi-model mean patterns of wetting and drying, with the high latitudes and the equatorial Pacific seeing increases, and the semi-arid regions of the Mediterranean, Australia and South Africa expecting further drying. As was the case for CMIP5 and previous multi-model ensembles, the average response across models is very robust to changes in the size and trajectory of well-mixed GHG
forcings, and therefore similar across scenarios. However, individual models' regional behavior may deviate from the average behavior significantly, especially in the regions of high internal variability, at the edges of sea-ice melt for temperature and in the equatorial pacific for precipitation.
The availability of ten (or more) ensemble members under SSP3-7.0 prescribed under Tier 2 and completed by five models at the time of writing allows to detect a tendency to decreasing internal
variability on decadal scales over time for both temperature and precipitation in all models (we note that several models have voluntarily provided initial condition ensembles of various sizes under other scenarios, but we have used one member only for those, which was all it was required for participation in ScenarioMIP). When considering annual frequencies only two of the models show significantly decreasing spread, and only for global temperature. The decadal scale results appear at odds with recent
studies that detected increased variability of precipitation with warming (Pendergrass et al., 2017; Yun et al., 2020), and call for in-depth studies of the sources and robustness of the behavior here described. For several pairs of models, ensemble spreads based on annual values turn out to be indistinguishable, while after computing running decadal means all models show significantly different spreads from one another, confirming that the representation of the climate system internal noise characteristics remains
model dependent (Parsons et al., 2020). CanESM5 provides 50 members and a subsampling of its ensemble confirms that ten realizations are sufficient to robustly estimate the forced signal of global temperature and precipitation by their averages, consistently with studies that have recently sought to investigate the question of how large a large ensemble needs to be for such estimation in those quantities (Milinski et al., 2020).
Lastly, a new feature of ScenarioMIP's design builds on the matrix framework combining SSPs to different radiative forcing levels and therefore allows estimates of the benefits of mitigation for two pairs of scenarios, one pair under SSP4, the other under SSP5, and also an evaluation of the path dependency of warming in the presence of an overshoot. The comparison of SSP5-8.5 to the overshoot pathway that departs from it in 2040 to strongly mitigate radiative forcing down to 3.4Wm$^{-2}$ by 2100
(SSP5-3.4OS) shows that the warming and absolute changes in precipitation avoided through late mitigation in 2040 could be up to half the expected changes under the high scenarios at the end of the century. The comparison of the other pair, SSP4-6.0 and SSP4-3.4 shows a similar geography of avoided physical impacts, but with smaller absolute differences, given the smaller reduction in radiative forcing between these two scenarios. We also compare the end points of SSP4-3.4, which follows
monotonically increasing forcing over the century, and of SSP5-3.4OS which overshoots the late century levels in radiative forcing and temperature, and therefore reaches them from above. Both temperature and precipitation changes (averaged over the last 20 years of the 21$^{st}$ Century) appear

comparable in magnitude, suggesting a short memory of the climate system with regard to global average temperature and precipitation, at least after it exceeds the ultimate target for up to 5 decades,
and by about 15°C of cumulative differential warming, as in this comparison. We note however that other environmental dimensions of climate change (such as ocean acidification or sea level rise) are not as easily reversible, if at all (Tokarska et al., 2019; Schwinger and Tjiputra, 2018; John et al., 2015; Mathesius et al, 2015; MacDougall et al., 2015; Tokarska and Zickfeld, 2015).
A more general analysis of the time it takes for the various scenarios to see a significant separation of
GSAT trajectories shows that the ensemble averages can show the climatological effects of mitigation (which we define as a persistent difference of at least a tenth of a degree) already within 15 years from the divergence of forcings when comparing SSP5-8.5 to the two lower scenarios, SSP1-1.9 and SSP1-2.6. "Adjacent" scenarios take longer to separate but they all do so, according to the ensemble means, by the mid 2040s. Individual pairs of trajectories from the ensemble members can take between about 5
and 20 years longer than the ensemble means (the longer period corresponding to the comparison between the two higher scenarios, SSP3-7.0 and SSP5-8.5).
We have limited this analysis to two variables and simple descriptive statistics of their behavior. The ScenarioMIP design together with the presence of complementary experiments in several other MIPs, and of the richness of the archived data (Jukes et al., 2020) from the ESMs simulations is going to
provide the basis for many more in-depth analyses of the physical system behavior. This will be further supported by a subset of CMIP6 models that are running CMIP5 RCPs, thus enabling a rigorous separation of the sources of variation between the two generations of experiments. Importantly, the ScenarioMIP effort aims at supporting integrated analyses of Earth and human systems' responses to future changes. These studies will integrate socio-economic changes described by SSPs with climate
system changes characterized by ESM outcomes to assess risks and possible mitigation and adaptation response options. While we don't address the integration of ScenarioMIP outcomes in interdisciplinary studies within this overview, that integration remains the overarching motivation for ScenarioMIP coordinated effort.


## Appendix: Additional Tables and Figures

**Table A1. Modeling centers and their model(s) contributing to CMIP6 ScenarioMIP. The citations are included in the main bibliography. DOIs refer to the data available through the Earth System Grid Federation. The last columns details the**
**experiments to which the model(s) contributed.**

| Institution | Model(s) | Model References Dataset DOIs | Experiments |
|---|---|---|---|
| Alfred Wegener Institute, Helmholtz Centre for Polar and Marine Research (Germany) | AWI | Semmler et al. (2020)  https://doi.org/10.22033/esgf/cmip6.376 | historical, ssp126, ssp245, ssp370, ssp585 |

| | | https://doi.org/10.22033/esgf/cmip6.359 | |
|---|---|---|---|
| Beijing Climate Center (China) | BCC-CSM2-MR | Wu et al. (2019)<br>Xin et al. (2019)<br><br>https://doi.org/10.22033/ESGF/C\MIP6.1732 | historical, ssp126, ssp245, ssp370, ssp585 |
| Canadian Centre for Climate Modelling and Analysis(Canada) | CanESM5-CanOE; CanESM5 | Swart et al. (2019)<br><br>https://doi.org/10.22033/ESGF/CMIP6.1317<br>https://doi.org/10.22033/ESGF/CMIP6.10207 | CanESM5-CanOE: historical, ssp126, ssp245, ssp370, ssp585<br><br>CanESM5: historical, ssp119, ssp126, ssp245, ssp370*, ssp434, ssp460, ssp534-over, ssp585 |
| Centre for Climate Change Research, Indian Institute of Tropical Meteorology (India) | IITM-ESM | Swapna et al. (2018)<br><br>https://doi.org/10.22033/ESGF/CMIP6.44 | historical, ssp126, ssp370, ssp585 |
| Centro Euro-Mediterraneo sui Cambiamenti Climatici (Italy) | CMCC-CM2-SR5 | Cherchi et al. (2019)<br>http://doi.org/10.22033/ESGF/CMIP6.3825<br>https://doi.org/10.22033/ESGF/CMIP6.3887<br>https://doi.org/10.22033/ESGF/CMIP6.3889<br>http://doi.org/10.22033/ESGF/CMIP6.3890<br>http://doi.org/10.22033/ESGF/CMIP6.3896 | historical, ssp126, ssp245, ssp370, ssp585 |
| Chinese Academy of Meteorological Sciences (China) | CAMS-CSM1.0 | Rong et al. (2018)<br>https://doi.org/10.22033/ESGF/CMIP6.11004 | historical, ssp119, ssp126, ssp245, ssp370, ssp585 |
| CNRM-CERFACS (France) | CNRM-CM6.1-HR; CNRM-CM6.1; CNRM-ESM2.1 | Roehrig et al. (2020)<br>Michou, M., et al. (2020)<br>Voldoire A., et al. (2019)<br>Seferian R. et al. (2019)<br><br>https://doi.org/10.22033/ESGF/CMIP6.4191, 2019.<br>https://doi.org/10.22033/ESGF/CMIP6.4197<br>https://doi.org/10.22033/ESGF/CMIP6.4198 | CNRM-CM6.1-HR: historical, ssp126, ssp245, ssp370, ssp585<br><br>CNRM-CM6.1: historical, ssp126, ssp245, ssp370, ssp585<br><br>CNRM-ESM2.1: historical, ssp119, ssp126, ssp245, ssp370, ssp434, |

| | | | ssp460, ssp534-over, ssp585 |
|---|---|---|---|
| CSIRO (Australia) | ACCESS-ESM1.5 | Ziehn et al. (2020)<br><br>https://doi.org/10.22033/ESGF/CMIP6.2291 | historical, ssp126, ssp245, ssp370, ssp585 |
| CSIRO-ARCCSS (Australia) | ACCESS-CM2 | Bi et al. ( 2020)<br><br>https://doi.org/10.22033/ESGF/CMIP6.2285 | historical, ssp126, ssp245, ssp370, ssp585 |
| EC-Earth Consortium | EC-Earth3, EC-Earth3-Veg | Doescher et al. (2020)<br><br>https://doi.org/10.22033/ESGF/CMIP6.727 | Both: historical, ssp119, ssp126, ssp245, ssp370, ssp585 |
| Department of Energy (USA) | E3SM-1.1 | Golaz et al. (2018), Burrows et al., (2020).<br><br>https://doi.org/10.22033/ESGF/CMIP6.4497<br>https://doi.org/10.22033/ESGF/CMIP6.15179 | Historical, ssp585 |
| First Institute of Oceanography (China) | FIO-ESM-2.0 | Bao et al. (2020)<br><br>https://doi.org/10.22033/ESGF/CMIP6.9208<br>https://doi.org/10.22033/ESGF/CMIP6.9209<br>https://doi.org/10.22033/ESGF/CMIP6.9214 | historical, ssp126, ssp245, ssp585 |
| Institut Pierre-Simon Laplace (France) | IPSL-CM6A-LR | Boucher. et al. (2020) Hourdin et al. (2019) Lurton et al. (2019)<br><br>https://doi.org/10.22033/ESGF/CMIP6.1532 | historical, ssp119, ssp126, ssp245, ssp370*, ssp434, ssp460, ssp534-over, ssp585 |
| Institute for Numerical Mathematic (Russia) | INM-CM5.0;INM-CM4.8 | Volodin et al. (2017) Volodin et al. (2018)<br><br>https://doi.org/10.22033/ESGF/CMIP6.12321<br>https://doi.org/10.22033/ESGF/CMIP6.12322 | Both: historical, ssp126, ssp245, ssp370, ssp585 |
| Institute of Atmospheric Physics (China) | FGOALS-f3-L;FGOALS-g3 | He et al. (2019) Li et al. (2019) Bao and Li. (2020)<br><br>https://doi.org/10.22033/ESGF/CMIP6.2046<br>https://doi.org/10.22033/ESGF/CMIP6.2056 | FGOALS-f3-L: historical, ssp126, ssp245, ssp370, ssp585<br><br>FGOALS-g3: historical, ssp126, ssp245, ssp370, ssp434, ssp534-over, ssp460, ssp585 |

| | | | |
|---|---|---|---|
| JAMSTEC, NIES,AORI, U. of Tokyo(Japan) | MIROC6; MIROC-ES2L | Tatebe et al ( 2019) Hajima  et al. (2020)<br><br>https://doi.org/10.22033/ESGF/CMIP6.898, 2019.<br>.<br>https://doi.org/10.22033/ESGF/CMIP6.936, 2019. | MIROC6:<br>historical, ssp119, ssp126, ssp245, ssp370, ssp434, ssp460, ssp534-over, ssp585<br><br>MIROC-ES2L:<br>historical, ssp119, ssp126, ssp245, ssp370, ssp534-over, ssp585 |
| Korea Institute of Ocean Science and Technology | KIOST-ESM | Kim et al., 2020<br><br>https://doi.org/10.22033/ESGF/CMIP6.1922<br><br>https://doi.org/10.22033/ESGF/CMIP6.11241 . | historical, ssp126, ssp245, ssp585 |
| Max Planck Institute for Meteorology (Germany), also Deutsches Klimarechenzentrum (Germany)  and Deutscher Wetterdienst (Germany) | MPI-ESM1.2-LR | Mauritsen  et al. (2019), Mueller  et al. (2018)<br><br>https://doi.org/10.22033/ESGF/CMIP6.2450<br>https://doi.org/10.22033/ESGF/CMIP6.1869<br>https://doi.org/10.22033/ESGF/CMIP6.793 | historical, ssp126, ssp245, sp370*, ssp585 |
| Met Office Hadley Center (UK) and Natural Environment Research Council (UK) | UKESM1.0-LL; HadGEM3-GC31-LL; HadGEM3-GC31-MM | Sellar et al (2019) Kuhlbrodt et al (2018) Williams et al (2017)<br><br>https://doi.org/10.22033/ESGF/CMIP6.1567<br>https://doi.org/10.22033/ESGF/CMIP6.10845 | UKESM1.0-LL:<br>historical, ssp119, ssp126, ssp245, ssp370, ssp534-over, ssp585<br><br>HadGEM3-GC31-LL:<br>historical, ssp126, ssp245, ssp585<br><br>HadGEM3-GC31-MM:<br>historical, ssp126, ssp585 |
| Meteorological Research Institute (Japan) | MRI-ESM2.0 | Yukimoto et al. (2019)<br><br>https://doi.org/10.22033/ESGF/CMIP6.638 | historical, ssp119, ssp126, ssp245, ssp370, ssp434, ssp460, ssp534-over, ssp585 |
| NASA GISS (USA) | GISS-E2.1-G | Kelley et al. (2020) Miller et al. (2020)<br><br>https://doi.org/10.22033/ESGF/CMIP6.2074 | historical, ssp126, ssp370, ssp434, ssp460, ssp585 |
| Nanjing University of Information Science and Technology (China) | NESM3 | Cao  et al. (2019) https://doi.org/10.22033/ESGF/CMIP6.2027 | historical, ssp126, ssp245, ssp585 |
| National Center for Atmospheric | CESM2(CAM6) and CESM2 | Danabasoglu  et al. (2019) | CESM2: |

| | | | |
|---|---|---|---|
| Research (USA) | (WACCM6) | https://doi.org/10.22033/ESGF/CMIP6.10026 https://doi.org/10.22033/ESGF/CMIP6.2201 | historical, ssp126, ssp245, ssp370 ssp585 CESM2 -WACCM: historical, ssp126, ssp245, ssp370, ssp534-over, ssp585 |
| National Institute of Meteorological Sciences, Korea Meteorological Administration (South Korea) | K-ACE-1-0-G | Lee et al. (2020) https://doi.org/10.22033/ESGF/CMIP6.2241 | historical, ssp126, ssp245, ssp370, ssp585 |
| NOAA-Geophysical Fluid Dynamics Laboratory (USA) | GFDL-CM4; GFDL-ESM4 | Held et al. (2019) Dunne et al. (2020) https://doi.org/10.22033/ESGF/CMIP6.1414 https://doi.org/10.22033/ESGF/CMIP6.9242. | GFDL-CM4: historical, ssp245, ssp585 GFDL-ESM4: historical, ssp119, ssp126, ssp245, ssp370, ssp585 |
| Norwegian Climate Center (Norway) | NorESM2-LM; NorESM2-MM; | Seland et al. (2020) Tjiputra et al. (2020) Counillon et al. (2016) https://doi.org/10.22033/ESGF/CMIP6.604 https://doi.org/10.22033/ESGF/CMIP6.608 https://doi.org/10.22033/ESGF/CMIP6.10894 | Both: historical, ssp126, ssp245, ssp370 ssp585 |
| University of Arizona (USA) | MCM-UA-1-0 | Delworth et al., (2002) Beadling et al. (2020) https://doi.org/10.22033/ESGF/CMIP6.2421 | historical, ssp126, ssp245, ssp370 ssp585 |

**Table A2: Modeling centers participating in CMIP5 and their models used in the comparison of SSPs and RCPs.**


| | |
|---|---|
| Beijing Climate Center (China) | BCC-CSM1-1; BCC-CSM1-1-M |
| BNU (China) | BNU-ESM |

| | |
|---|---|
| Canadian Centre for Climate Modelling and Analysis(Canada) | CanESM2 |
| CNRM-Cerfacs (France) | CNRM-CM5 |
| CSIRO-BOM (Australia) | ACCESS1-0;ACCESS1-3; CSIRO-Mk3-6-0 |
| EC-Earth Consortium | EC-Earth |
| Euro-Mediterranean Center on Climate Change (Italy) | CMCC-CM;CMCC-CMS |
| First Institute of Oceanography (China) | FIO-ESM |
| Institut Pierre Simon Laplace (France) | IPSL-CM5A-LR;IPSL-CM5A-MR;IPSL-CM5B-LR |
| Institute for Numerical Mathematic (Russia) | INM-CM4 |
| Institute of Atmospheric Physics (China) | FGOALS-g2 |
| JAMSTEC, NIES, CCSR, U. of Tokyo(Japan) | MIROC-ESM; MIROC-ESM-CHEM;MIROC5 |
| Max Planck Institute (Germany) | MPI-ESM-LR; MPI-ESM-HR |
| Met Office Hadley Center (UK) | HadGEM2-AO; HadGEM2-CC; HadGEM2-ES |
| Meteorological Research Institute (Japan) | MRI-CGCM3 |
| NASA GISS (USA) | GISS-E2-R; GISS-E2-R-CC;GISS-E2-H; GISS-E2-H-CC |
| National Center for Atmospheric Research (USA) | CCSM4; CESM1-BGC; CESM1-CAM5; CESM1-WACCM |
| NOAA-Geophysical Fluid Dynamics Laboratory (USA) | GFDL-CM3; GFDL-ESM2G;GFDL-ESM2M |
| Norwegian Climate Center (Norway) | NorESM1-ME;NorESM1-M |

**Table A3: CMIP6 models' projected warming under the five scenarios by 2041-2060 and 2081-2100 relative to the historical baseline of 1995-2014. Ensemble mean values and, in square brackets, 5-95% confidence intervals (+/- 1.64σ).**

| Surface Air Temperature Change (°C) (1995-2014) | SSP1-1.9 (13 models) | SSP1-2.6 (38 models) | SSP2-4.5 (37 models) | SSP3-7.0 (33 models) | SSP5-8.5 (39 models for global, 38 for land/ocean) |
|---|---|---|---|---|---|
| 2041-2060 Global | 0.83 [0.31,1.36] | 1.07 [0.51,1.63] | 1.32 [0.77,1.88] | 1.46 [0.82,2.10] | 1.74 [1.05,2.42] |
| 2081-2100 Global | 0.69 [0.13,1.25] | 1.23 [0.40,2.05] | 2.14 [1.27,3.00] | 3.16 [1.95,4.38] | 3.99 [2.40,5.57] |

| 2041-2060 Land-Only | 1.16 [0.45,1.87] | 1.45 [0.73,2.17] | 1.80 [1.04,2.55] | 1.97 [1.10,2.84] | 2.35 [1.43,3.28] |
|---|---|---|---|---|---|
| 2081-2100 Land-Only | 0.96 [0.17,1.74] | 1.61 [0.56,2.65] | 2.85 [1.73,3.97] | 4.26 [2.63,5.90] | 5.46 [3.36,7.57] |
| 2041-2060 Ocean-Only | 0.69 [0.26,1.12] | 0.91 [0.42,1.40] | 1.12 [0.64,1.59] | 1.24 [0.70,1.78] | 1.46 [0.87,2.05] |
| 2081-2100 Ocean-Only | 0.57 [0.11,1.03] | 1.06 [0.33,1.79] | 1.83 [1.07,2.59] | 2.70 [1.65,3.74] | 3.41 [2.06,4.75] |

**Table A4: CMIP6 models' projected changes in precipitation under the five scenarios by 2041-2060 and 2081-2100 expressed as percentages relative to the historical baseline of 1995-2014. Ensemble mean values and, in square brackets, 5-95% confidence intervals (+/- 1.64σ).**

| Precipitation Change (%) (1995-2014) | SSP1-1.9 (13 models) | SSP1-2.6 (37 models) | SSP2-4.5 (36 models) | SSP3-7.0 (33 models) | SSP5-8.5 (38 models for global, 37 for land/ocean) |
|---|---|---|---|---|---|

| | | | | |
|---|---|---|---|---|
| 2041-2060 Global | 2.04 [0.53,3.56] | 2.37 [0.63,4.10] | 2.33 [0.81,3.85] | 2.08 [0.58,3.57] | 2.78 [0.89,4.67] |
| 2081-2100 Global | 2.02 [0.37,3.67] | 3.05 [0.81,5.28] | 4.19 [1.79,6.59] | 4.88 [1.92,7.85] | 7.30 [-0.65,15.26] |
| 2041-2060 Land-Only | 2.59 [0.53,4.66] | 2.90 [-0.07,5.87] | 2.91 [0.21,5.61] | 2.67 [-0.22,5.57] | 3.90 [0.55,7.24] |
| 2081-2100 Land-Only | 2.32 [0.03,4.61] | 3.57 [0.04,7.11] | 4.83 [1.06,8.60] | 6.19 [1.14,11.24] | 8.61 [2.37,14.85] |
| 2041-2060 Ocean-Only | 1.81 [0.44,3.17] | 2.21 [0.54,3.88] | 2.16 [0.60,3.72] | 1.95 [0.53,3.38] | 2.56 [0.72,4.40] |
| 2081-2100 Ocean-Only | 1.87 [0.36,3.38] | 2.88 [0.77,5.00] | 4.01 [1.55,6.47] | 4.67 [1.62,7.72] | 6.21 [2.07,10.35] |

**Table A5: Time of separation between smoothed GSAT trajectories under pairs of scenarios. Shown is the year by which the ensemble means separate, and, in square brackets, the year by which the last of the separation among individual models' trajectories takes place. Separation is defined as the emergence of a positive difference (we use 0.1°C as threshold) that persists for the remainder of the century. We first apply a 21-year running mean to the GSAT time series in order to characterize separation "of climates".**

| | SSP1-2.6 | SSP2-4.5 | SSP3-7.0 | SSP5-8.5 |
|---|---|---|---|---|
| **SSP1-1.9** | 2042 [2050] | 2034 [2043] | 2031 [2041] | 2027 [2036] |
| **SSP1-2.6** | | 2039 [2053] | 2037 [2048] | 2030 [2036] |
| **SSP2-4.5** | | | 2046 [2058] | 2031 [2044] |
| **SSP3-7.0** | | | | 2034 [2053] |

**Table A6: Projected warming and precipitation change under comparable scenarios, for CMIP5 and CMIP6 ensembles, and for the CMIP6 ensemble constrained by the method of Tokarska et al. (2020). For the latter the number of models remains the same as for the unconstrained projections. Differently from Tables A3 and A4, which use the CMIP6 current baseline period of 1995-2014, here all changes are relative to the CMIP5 current baseline period of 1986-2005. See also Fig. A6 for a graphical representation of the raw and constrained temperature projections for 2081-2100, besides Fig. 4 in the main text.**

| GSAT Change (°C) (1986-2005) | | | Precipitation Change (%) (1986-2005) | | |
|---|---|---|---|---|---|
| | 2041-2060 | 2081-2100 | | 2041-2060 | 2081-2100 |
| RCP 2.6 (28 models) | 1.01 (0.50,1.62) | 1.01 (0.23,1.74) | RCP 2.6 (27 models) | 2.20 (0.90,3.50) | 2.52 (0.77,4.27) |
| RCP 2.6 constrained | 0.85 (0.38,1.31) | 0.83 (0.15,1.50) | | | |
| SSP1-2.6 (37 models) | 1.35 (0.77,2.06) | 1.47 (0.80,2.44) | SSP1-2.6 (37 models) | 2.78 (0.95,4.61) | 3.46 (1.14,5.79) |
| SSP1-2.6 constrained | 1.07 (0.54,1.59) | 1.12 (0.38,1.85) | | | |
| RCP 4.5 (36 models) | 1.33 (0.86,1.83) | 1.90 (1.07,2.72) | RCP4.5 (38 models) | 2.42 (1.23,3.61) | 3.64 (1.71,5.57) |
| RCP 4.5 constrained | 1.19 (0.75,1.62) | 1.71 (0.87, 2.56) | | | |
| SSP2-4.5 (38 models) | 1.57 (1.04,2.30) | 2.39 (1.53,3.50) | SSP2-4.5 (36 models) | 2.75 (1.11,4.39) | 4.62 (2.08,7.16) |
| SSP2-4.5 constrained | 1.30 (0.80,1.79) | 2.00 (1.20,2.80) | | | |
| RCP 8.5 (37 models) | 1.79 (1.25,2.37) | 3.71 (2.71,4.71) | RCP8.5 (36 models) | 3.00 (1.54,4.46) | 6.20 (3.35,9.06) |
| RCP 8.5 constrained | 1.62 (1.12,2.12) | 3.45 (2.43,4.46) | | | |
| SSP5-8.5 (40 models) | 2.02 (1.37,2.95) | 4.38 (2.92,6.20) | SSP5-8.5 (37 models) | 3.25 (1.26,5.24) | 7.05 (3.03,11.06) |
| SSP5-8.5 constrained | 1.62 (0.99,2.24) | 3.60 (2.13,5.05) | | | |

940

**Table A7: Like Table 1 in the main text, times of crossing of different warming levels by the available ensembles running the various scenarios (best estimate and range - in square brackets - based on the 5-95% range of the ensemble after smoothing the trajectories by eleven-year running means). Crossing of these levels are defined by using anomalies wrt 1995-2014 for the model ensembles, and adding the offset of 0.84°C to derive warming from pre-**
945 **industrial.**

Since the number of models available under each scenario varies, and in some cases not all models reach a given warming level, those numbers are shown in parentheses. However, the estimates are based on the ensemble means and ranges computed from the whole ensemble, not just from the models that reach a given level. An estimate marked as NA is to be interpreted as "not reaching a given level by 2100". In cases where the ensemble average remains below the warming level for the whole century, it is possible for the central estimate to be NA, while the earlier time of the confidence interval is not, since the upper bound of the ensemble range may still reach that warming level.

|  | SSP1-1.9 | SSP1-2.6 | SSP2-4.5 | SSP3-7.0 | SSP5-8.5 |
|---|---|---|---|---|---|
| **1.5°C** | 2029 [2021,NA] (11/13) | 2028 [2020,NA] (37/38) | 2027 [2020,2046] (37/37) | 2028 [2020,2045] (33/33) | 2026 [2020,2040] (39/39) |
| **2.0°C** | NA [2036,NA] (2/13) | 2060 [2032,NA] (21/38) | 2045 [2031,2077] (37/37) | 2043 [2031,2064] (33/33) | 2039 [2030,2054] (39/39) |
| **3.0°C** | NA [NA,NA] (0/13) | NA [NA,NA] (1/38) | 2092 [2059,NA] (21/37) | 2068 [2052,NA] (33/33) | 2059 [2047,2082] (39/39) |
| **4.0°C** | NA [NA,NA] (0/13) | NA [NA,NA] (0/38) | NA [NA,NA] (2/37) | 2091 [2071,NA] (18/33) | 2077 [2062,NA] (33/39) |
| **5.0°C** | NA [NA,NA] (0/13) | NA [NA,NA] (0/38) | NA [NA,NA] (0/37) | NA [2088,NA] (4/33) | 2094 [2074,NA] (21/39) |

**Table A8: Warming level crossings for CMIP5 and CMIP6 scenarios/ensembles. Shown are times when an 11-year running average of the ensemble mean trajectory, and the lower and upper bounds of its 90% confidence interval (1.64σ, where σ is the ensemble standard deviation after smoothing) cross various warming levels, under the three comparable scenarios: SSP1-2.6, SSP2-4.5 and SSP5-8.5 for CMIP6 models, RCP2.6, RCP4.5 and RCP8.5 for CMIP5 models. NAs values indicate that the corresponding ensemble metric (mean, lower or upper bound of the confidence**

interval) does not reach the corresponding warming level by 2100. The numbers on the bottom row of each cell
indicate the number of models that reach that warming level. The largest ensemble available under all three scenarios
considered is used in both cases, with 36 CMIP6 models and 29 CMIP5 models.

|  | SSP1-2.6 | SSP2-4.5 | SSP5-8.5 | RCP2.6 | RCP4.5 | RCP8.5 |
|---|---|---|---|---|---|---|
| **1.5°C** | 2025 (2020,NA) 35/36 | 2026 (2020,2047) 36/36 | 2024 (2020,2040) 36/36 | 2034 (2018,NA) 23/29 | 2029 (2021,2055) 29/29 | 2027 (2018,2039) 29/29 |
| **2.0°C** | 2056 (2029,NA) 18/36 | 2043 (2028,2080) 36/36 | 2038 (2027,2054) 36/36 | NA (2040,NA) 7/29 | 2051 (2035,NA) 24/29 | 2041 (2030,2056) 29/29 |
| **3.0°C** | NA (2092,NA) 2/36 | 2089 (2055,NA) 20/36 | 2058 (2045,2082) 36/36 | NA (NA,NA) 0/29 | NA (2069,NA) 7/29 | 2063 (2051,2085) 29/29 |
| **4.0°C** | NA (NA,NA) 0/36 | NA (2092,NA) 2/36 | 2076 (2060,NA) 31/36 | NA (NA,NA) 0/29 | NA (NA,NA) 0/29 | 2084 (2068,NA) 24/29 |
| **5.0°C** | NA (NA,NA) 0/36 | NA (NA,NA) 0/36 | 2093 (2073,NA) 20/36 | NA (NA,NA) 0/29 | NA (NA,NA) 0/29 | NA (2083,NA) 10/29 |

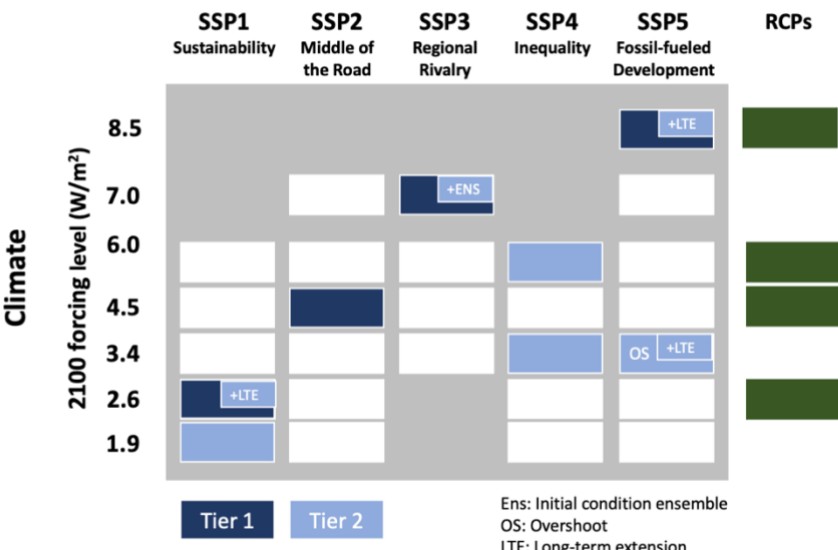

**Figure A1: ScenarioMIP design (modified from O'Neill et al., 2020). White and colored boxes indicate achievable 2100 levels of forcings under the different SSPs. Grey areas are at the intersection of SSPs and radiative forcing levels that were not achievable by any of the IAMs employed to produce these scenarios.**

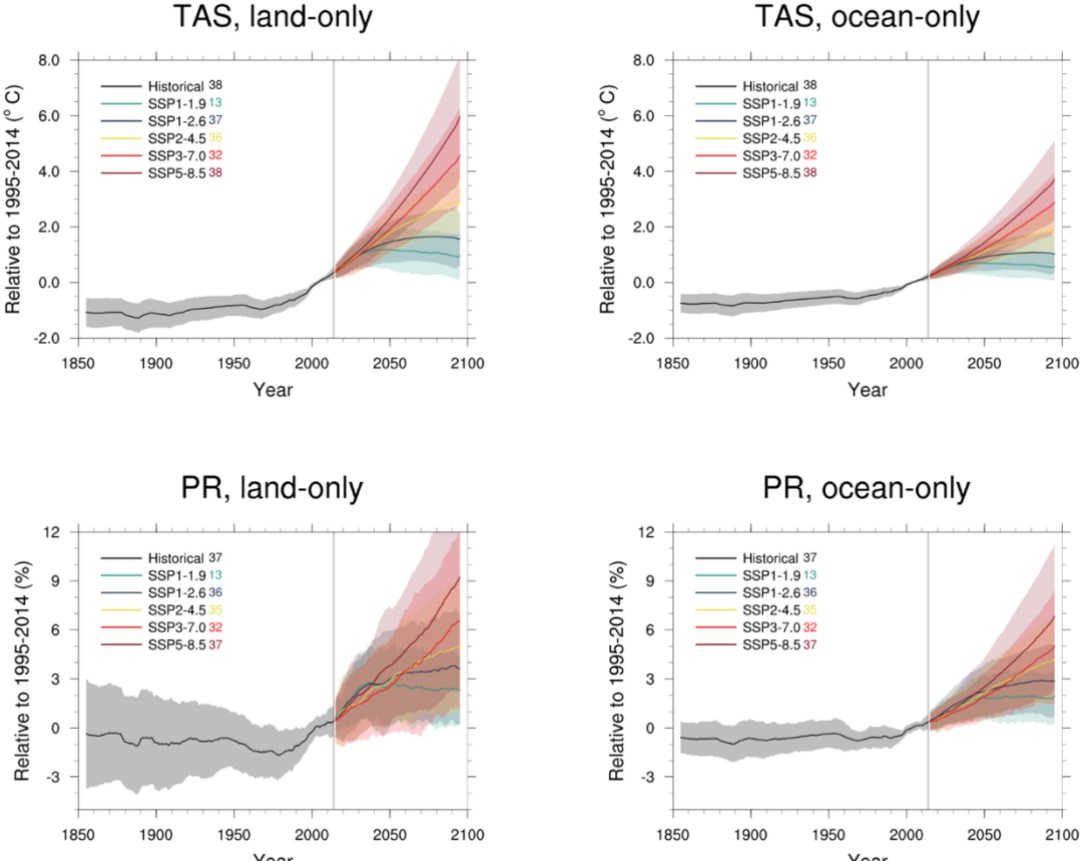

1000 **Figure A2: Land-only and ocean-only average time series of temperature and percent precipitation changes relative to 1995-2014, for the 4 scenarios of Tier 1, SSP1-2.6, SSP2-4.5, SSP3-7.0, SSP5-8.5, and SSP1-1.9.**

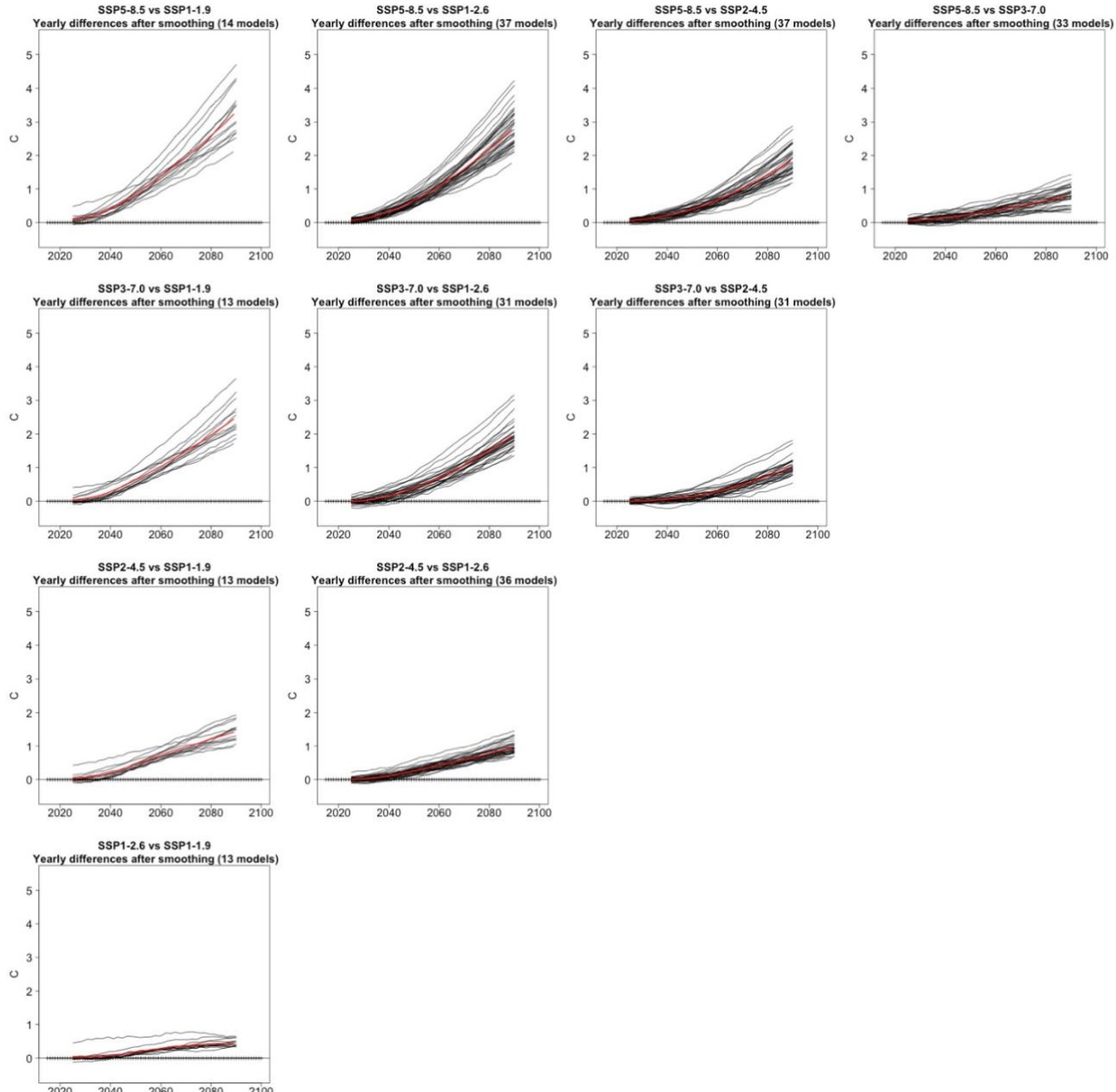

1005

**Figure A3: Time series of year-by-year differences in GSAT between each scenario run in Tier 1 and each of the lower scenario runs (including SSP1-1.9). The time series from the individual models were first smoothed by a 21-year running mean. First row: differences between SSP5-8.5 and, respectively, SSP1-1.9, SSP1-2.6, SSP2-4.5 and SSP3-7.0. Second row: differences between SSP3-7.0 and respectively SSP1-1.9, SSP1-2.6 and SSP2-4.5. Third row: differences between** 1010 **SSP2-4.5 and SSP1-1.9 and SSP1-2.6. Fourth row: differences between SSP1-2.6 and SSP1-1.9. Each black line corresponds to an individual model's time series of differences. The red line is the ensemble mean difference. The ensemble size varies across the plots based on the number of models available for which the difference can be computed. It is as small as 10 members for those**

**differences involving SSP1-1.9 and as large as 25 to 30 members when both scenarios belong to Tier 1.**

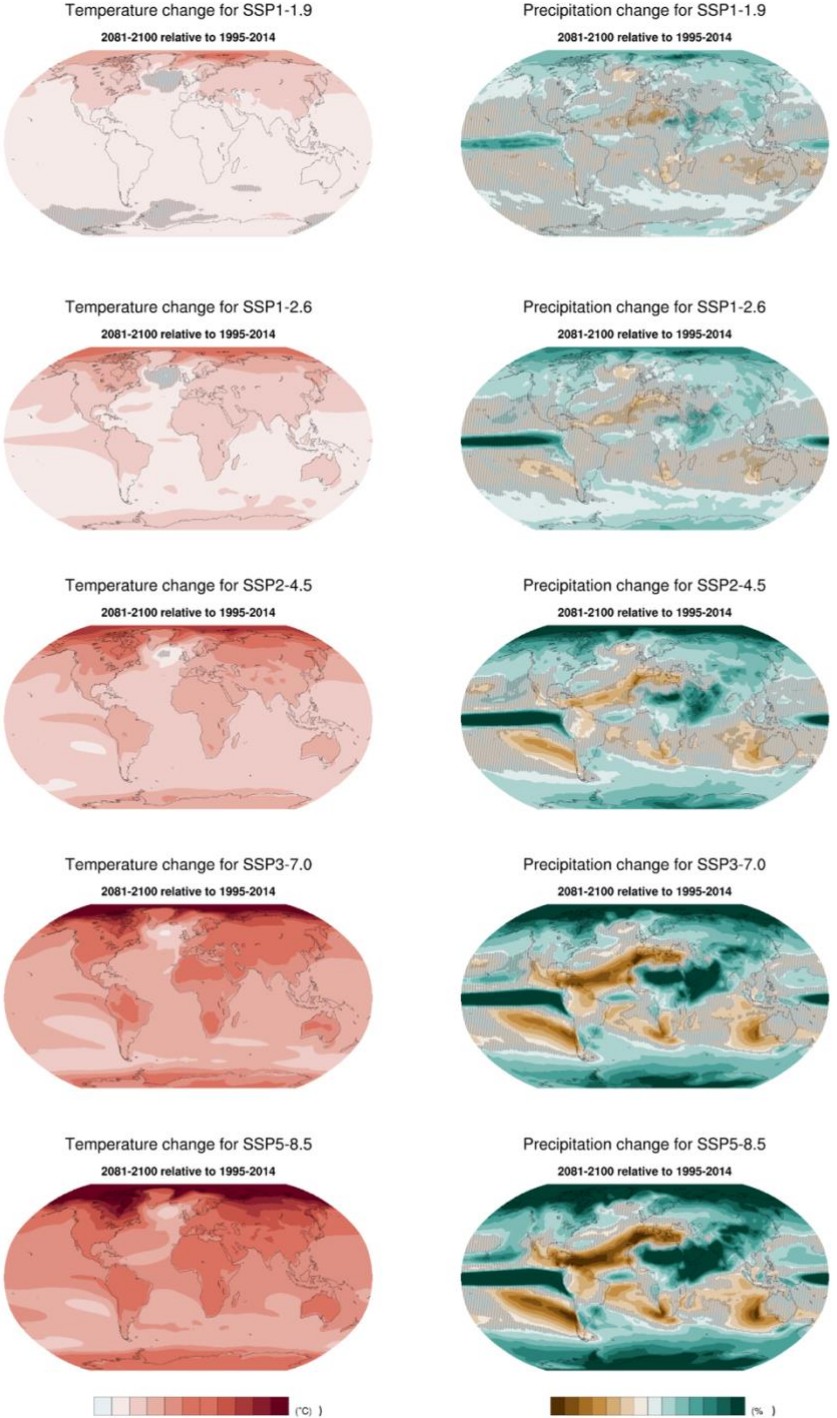

 **Figure A4: Patterns of changes by 2081-2100 relative to 1995-2014 in surface air temperature (°C) and precipitation (%) under the five scenarios. Stitched areas are not significant, i.e., the magnitude of the change does not exceed the models' standard deviation.**

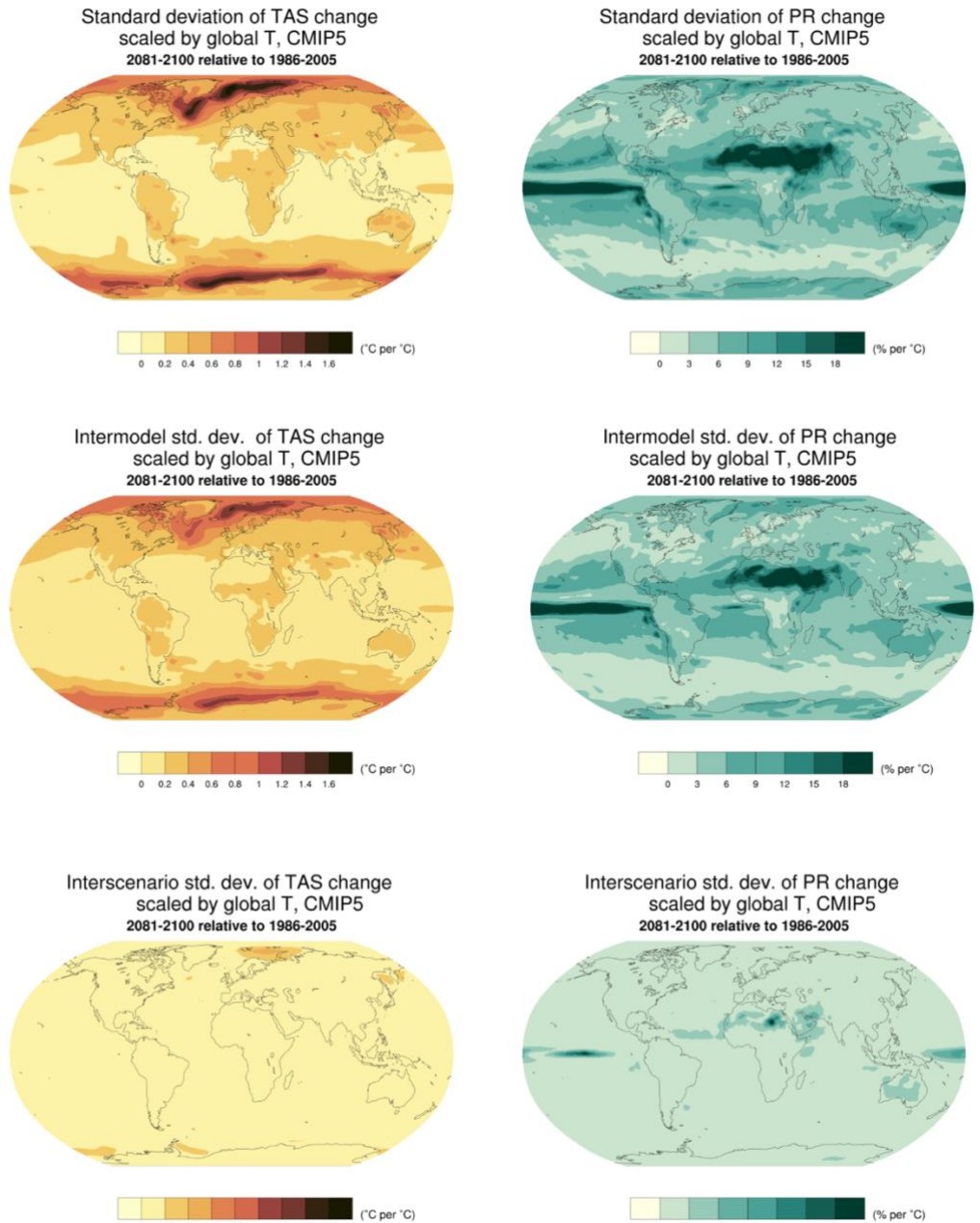

**Figure A5: Top row: standard deviation of normalized patterns for individual CMIP5 models and scenarios. The individual patterns are the elements from which the averages shown in Fig. 3 are computed. Center row: Standard deviation of normalized patterns, after averaging across scenarios, highlighting the role of inter-model variability. Bottom row: Standard deviation of normalized patterns after averaging across models, highlighting the role of inter-scenario variability. These standard deviations can be compared with the corresponding results from CMIP6 models/scenarios in Fig. A5.**

1025

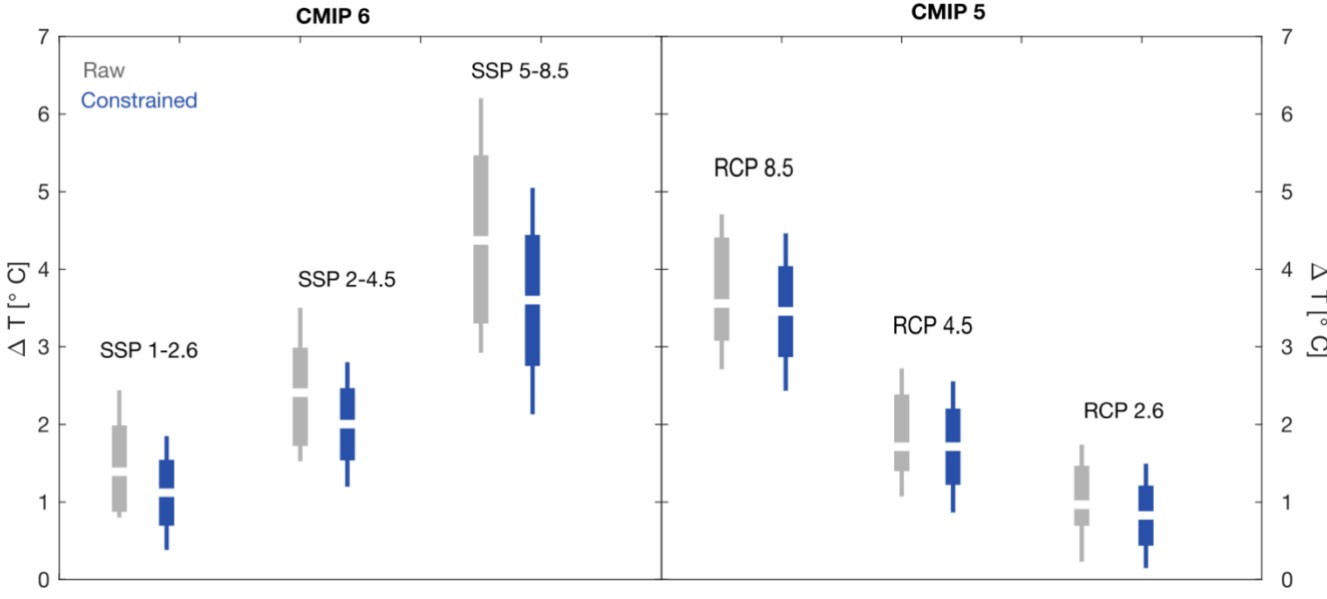

1030

**Figure A6: A closer look at the effects of applying the Tokarska et al. 2020 constraints to CMIP6 and CMIP6 projections (mean changes at 2081-2100 compared to 1986-2005) for the nominally corresponding scenarios.**

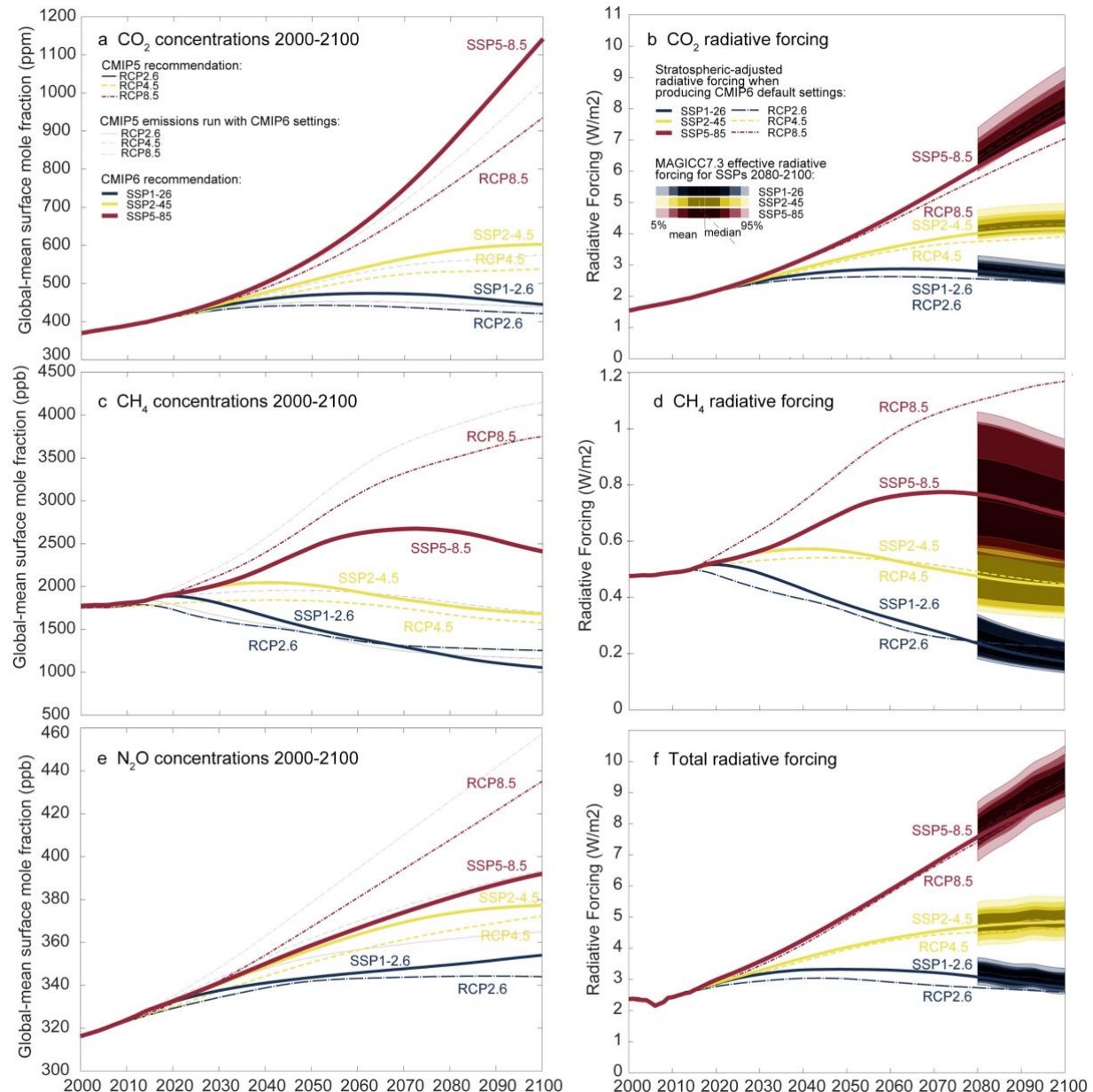

**Figure A7: Comparison of CO2, CH4 and N2O concentrations and radiative forcings for the concentration-driven CMIP5 runs with RCP-Y scenarios (Meinshausen et al., 2011) and CMIP6 runs with SSPX-Y scenarios (Meinshausen et al., 2020). The higher scenario SSP5-8.5 features higher CO2 concentrations largely due to updated carbon cycle settings. RCP8.5 emissions with the same carbon cycle settings (shown as thin dashed line in panel a) would produce similar CO2 concentrations. The methane and nitrous oxide concentrations are however lower in SSP5-8.5 than in RCP8.5 (despite updated gas cycles producing higher concentrations for the same emission trajectory). Panels a, c and e adapted from Fig. 11 in Meinshausen et al. 2020. At the time of producing the SSPs (March 2018), stratospheric-adjusted radiative forcings have been used to compare the nameplate radiative forcing levels in 2100 using MAGICC6.8 with IPCC AR5 consistent settings (see panels b, d, f). Effective radiative forcings (ERFs) take additional adjustments into account that are non-temperature induced and differ from stratospheric-adjusted radiative**

forcings. Shown are 2080-2100 probabilistic results of SSP ERFs, using MAGICC7.3. These ERFs differ from SARFs and tend to be higher for CO2 and total radiative forcings (see panel b and f). Given that the efficacy and rapid adjustments are different for different forcing agents, also the match between RCPs and SSP scenarios differs when comparing them in the effective radiative forcing space, rather than in terms of their stratospheric-adjusted radiative forcings.

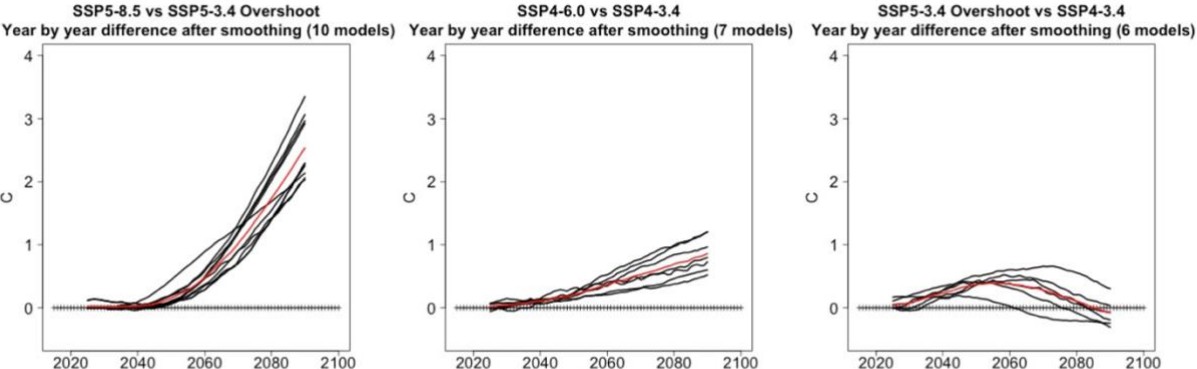

Figure A8: As in Fig. A3, year by year GSAT differences for the two pairs of scenarios differing only by the amount of mitigation assumed (left and center panel) and for the two scenarios that achieve the same level of radiative forcing by 2100, one by overshooting it in the middle of the century (right panel). From left to right: year by year differences for the seven models that ran SSP5-8.5 and SSP5-3.4OS, for the seven models that ran SSP4-6.0 and SSP4-3.4, and for the 5 models that ran SSP4-3.4 and SSP5-3.4OS. Black lines are differences computed between pairs of GSAT trajectories for each of the models. Red lines are differences between the two ensemble mean trajectories.

## Data and Code Availability

CMIP5 (see Table A2) and CMIP6 (see Table A1) model output is available through the Earth System Grid Foundation (ESGF) and can be directly used within the ESMValTool (e.g. https://esgf-data.dkrz.de/projects/esgf-dkrz/). The corresponding recipe that can be used to reproduce the figures of this paper will be included in ESMValTool v2.0 (Righi et al., 2020; Eyring et al., 2019a; Lauer et al., 2020; Weigel et al., 2020) as soon as the paper is published. The ESMValTool is released under the Apache License, VERSION 2.0. The ESMValTool code is available from the ESMValTool webpage at https://www.esmvaltool.org/ and from github (https://github.com/ESMValGroup/ESMValTool). As of December 2020, 27 modeling centers participated in ScenarioMIP by running at a minimum its Tier 1 experiments and provided their output through the ESGF. Table A1 lists them, together with their model(s) and the doi referencing the data.

## Author Contributions

C. Tebaldi, V. Eyring, J. Fyfe and E. Fischer designed and organized the analysis. K. Debeire performed data processing and analysis, and drew all figures and most of the tables. C. Tebaldi wrote

the first draft of the paper. All authors provided input, comments and editing on the various parts of the analysis. In addition, modeling centers representatives (from S. Bauer to T. Ziehn in the authors' list) were responsible for performing the ScenarioMIP simulations and publishing their model output to the ESGF. The authors declare that they have no conflict of interest.

**Competing Interests**

The authors declare that they have no conflict of interest.

**Acknowledgements**

C. Tebaldi was supported by the Energy Exascale Earth System Model (E3SM) project, funded by U.S. Department of Energy, Office of Science, Office of Biological and Environmental Research. The Pacific Northwest National Laboratory is operated by Battelle for the US Department of Energy under Contract DE-AC05-76RLO1830. This work has been also supported by the European Union's Horizon 2020 Framework Programme for Research and Innovation "Coordinated Research in Earth Systems and Climate: Experiments, kNowledge, Dissemination and Outreach (CRESCENDO)" project under Grant Agreement No. 641816, and the EVal4CMIP project funded by the Helmholtz Society. We acknowledge the World Climate Research Programme (WCRP), which, through its Working Group on Coupled Modelling, coordinated and promoted CMIP. We thank the climate modeling groups (listed in Tables A1 and A2) for producing and making available their model output, the Earth System Grid Federation (ESGF) for archiving the data and providing access, and the multiple funding agencies who support CMIP, ESGF and the individual modeling centers efforts. Work at LLNL was performed under the auspices of the U.S. Department of Energy by Lawrence Livermore National Laboratory under Contract DE-AC52-07NA27344. A. Voldoire and R. Seferian thank the H2020 CONSTRAIN under the grant agreement N. 820829 and the support of the team in charge of the CNRM-CM climate model. Supercomputing time was provided by the Meteo-France/DSI supercomputing center. The computational resources of the Deutsches Klimarechenzentrum (DKRZ, Hamburg, Germany) that allowed the analysis of this study with the ESMValTool are kindly acknowledged.

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
