# Peer review of "Intercomparison Project (ScenarioMIP) of CMIP6"

_Earth System Dynamics, 2020_

## Referee Comment (RC1) · Anonymous Referee #1 · 28 Oct 2020

This paper provides a first overview over temperature and precipitation projections from the ScenarioMIP simulations conducted for CMIP6. As such, it is more of a documentation than a cutting edge research paper, in part also because many of the CMIP6 results are very similar to CMIP5. However, I think this is ok given the clearly defined scope of the paper and the assumed goal to support the AR6 process.

The paper is well written with clear figures and sound methods. I have a few minor comments that could be considered before publication.

Minor comments:

L130-: is there a reason these papers aren't cited in the classic style with "XYZ et al."?

L147-: nothing wrong with this paragraph given the high level nature of the study, but I think it would still benefit the reader to end the introduction with a set of specific research questions that are being addressed in this paper.

L205: "likely conservative". I think I know what the authors are trying to say, but I still think it would be clearer to be explicit and say "underestimated" and maybe also briefly explain why.

Paragraph on L223: Possibly relevant paper, although conceptually similar to Tebaldi and Friedlingstein: Marotzke (2019).

Generally, figure labels are too small. White space between panels could be decreased to make maps bigger and better readable.

Fig A5 is insightful and given its relatively detailed discussion in the text, I suggest to add it to Fig 2.

L325: ok, but how about a simple difference maps of CMIP6-CMIP5?

Fig 4b: could the Tokarska constraint be applied to CMIP5 as well to show that it makes less of a difference in CMIP5 (as expected)?

L465: Semantics, but technically if the reference period is 1995-2014 and thus its mid year is 2004, then the models have 16 years, not 5, to reach 1.5°C.

L510: this is indeed new, to me at least. Doesn't look like there was such a coherent signal in CMIP5 models (Pendergrass et al. 2017). It is also a bit counterintuitive perhaps, but definitely plausible. A little more context on the history of this (why is the result new?) and maybe maps to indicate where this is most likely to originate from would be appreciated.

L520-: This recent paper could provide some more context here: Milinski et al. (2019)

The discussion section is a bit light on references to other papers. I don't want to provide a list, but I trust the authors are able to do a better job at this given their collective

expertise. One specific paper that is interesting but still relatively new: Parsons et al. (2020)

References

Marotzke, J., 2019: Quantifying the role of internal variability in the temperature we expect to observe in the coming decades. Wiley Interdiscip. Rev. Clim. Chang., 10, 1–12, https://doi.org/10.1002/wcc.563.

Milinski, S., N. Maher, and D. Olonscheck, 2019: How large does a large ensemble need to be? Earth Syst. Dyn. Discuss., 1–19.

Parsons, L. A., M. K. Brennan, R. C. J. Wills, and C. Proistosescu, 2020: Magnitudes and Spatial Patterns of Interdecadal Temperature Variability in CMIP6. Geophys. Res. Lett., 47, 1–11, https://doi.org/10.1029/2019GL086588.

Pendergrass, A. G., R. Knutti, F. Lehner, C. Deser, and B. M. Sanderson, 2017: Precipitation variability increases in a warmer climate. Sci. Rep., 7, 17966, https://doi.org/10.1038/s41598-017-17966-y.

---

## Referee Comment (RC2) · Anonymous Referee #2 · 14 Nov 2020

The authors present the results of CMIP6 ScenarioMIP focusing on surface air temperature and precipitation. They find similar characteristics in the temperature and precipitation patterns to those simulated by CMIP5 models. They also depict the temperature and precipitation changes under the new scenarios in CMIP6, such as SSP1-1.9, SSP3-7.0 and SSP5-3.4OS.

This is a comprehensive study of CMIP6 ScenarioMIP, summarizing the results of multiple global warming scenarios from CMIP6 and providing some new insights. However, I feel some clarifications are needed to justify the results.

Lines 212-213 "This suggests that the model response uncertainty increases for

stronger responses". Why does model response uncertainty increase for stronger responses?

Line 223: The authors define "separation", and use "a 21-year running mean" (Line 227) and "choose 0.1°C as the threshold" (Line 228). Why is a 21-year running mean used here instead of a 9-year or 11-year running mean? The latter two running means I can understand aim to minimize the interannual variability, but what is the purpose of the 21-year running mean? Will the results of "separation" be sensitive to the two parameters of the running mean and 0.1°C?

Fig. 1. Are they annual mean time series of temperature and precipitation?

Part 3.1.3 Would be it possible to show the difference between the SSPs and RCPs more explicitly? For example, plot the time series of the differences of temperature and precipitation between the SSPs and RCPs? Also show the spatial pattern of the difference, i.e., the global map of the difference of temperature and precipitation change/trend in the 21st century between the SSPs and RCPs? Besides, please explain why are SSPs induced changes different from their corresponding RCPs? What is the factor accounting for the difference and through what processes?

Additionally, please note the superscripts and subscripts in the denotations. For example, "2.6Wm-2 " (Line 364), "0.5 Wm-2" (Line 400), "$CO_2$, $CH_4$ and $N_2O$" (Line 419).

---

## Author Comment (AC1) · 2 Dec 2020

**Response to Referee #1 (in blue italics)**

This paper provides a first overview over temperature and precipitation projections from the ScenarioMIP simulations conducted for CMIP6. As such, it is more of a documentation than a cutting edge research paper, in part also because many of the CMIP6 results are very similar to CMIP5. However, I think this is ok given the clearly defined scope of the paper and the assumed goal to support the AR6 process.

*Thank you, indeed that is the nature of this overview paper, which also serves the purpose of involving and acknowledging the representatives of the modeling centers that participated in our MIP.*

The paper is well written with clear figures and sound methods.

*Thank you, we aim at providing all the scripts to reproduce the analysis and figures of the paper.*

I have a few minor comments that could be considered before publication.
Minor comments:
L130-: is there a reason these papers aren't cited in the classic style with "XYZ et al."?

*This was an oversight at least in part. A couple of links are meant to remain as such, because they are pointers to the ESGF inputs4mip downloadable data. The links to papers are indeed going to be substituted by citations according to the standard style in the final version, correcting the oversight.*

L147-: nothing wrong with this paragraph given the high level nature of the study, but I think it would still benefit the reader to end the introduction with a set of specific research questions that are being addressed in this paper.

*We agree and will reformulate the paragraph in the style suggested. It will read as:*
*"In this study, we focus the analysis on the future evolution of average temperatures and precipitation. We address questions regarding the strength of the signal under the different scenarios and compared to similar CMIP5 scenarios, the identification of the time of separation between the temperature trajectories under the different scenarios, and the time at which they cross global warming thresholds. We also analyze spatial patterns of change addressing questions of robustness between the CMIP5 and CMIP6 multi-model ensembles, and within the CMIP6 ensemble among models and scenarios."*

L205: "likely conservative". I think I know what the authors are trying to say, but I still think it would be clearer to be explicit and say "underestimated" and maybe also briefly explain why.

*Agreed. We will rephrase as in: "likely underestimated, given that we are using only one run per model, while larger initial condition ensembles would better characterize each model's internal variability."*

Paragraph on L223: Possibly relevant paper, although conceptually similar to Tebaldi and Friedlingstein: Marotzke (2019).

*Thank you, we will add this citation.*

Generally, figure labels are too small. White space between panels could be decreased to make maps bigger and better readable.

*We plan to redraw all figures updating the content as a few more models have contributed their output to the ESGF in the intervening time since the paper was submitted. We will make use of this suggestion as well while doing that.*

Fig A5 is insightful and given its relatively detailed discussion in the text, I suggest to add it to Fig 2.

*It was actually like that in a previous version and we will be happy to reinstate it!*

L325: ok, but how about a simple difference maps of CMIP6-CMIP5?

*Good point. We will add the two difference plots under Fig3.*

Fig 4b: could the Tokarska constraint be applied to CMIP5 as well to show that it makes less of a difference in CMIP5 (as expected)?

*We have been remiss in a thorough discussion of this result and we plan to expand on it in the next version. Katarzyna Tokarska is now a co-author, and we will show the result of applying the constraints also to CMIP5, placing the corresponding brackets on the right-hand plot of Figure 4. These will show that the effects of applying observational constraints to CMIP5 is significantly less consequential. We will discuss this in the text, where appropriate.*

L465: Semantics, but technically if the reference period is 1995-2014 and thus its mid year is 2004, then the models have 16 years, not 5, to reach 1.5_C.

*Good point. After reading that sentence again we decided to limit the parenthetical to pointing out the last year of the historical period, in order to avoid complications. We have however added some more caveats and discussion of these results, pointing at the possible discrepancies between observed and modeled trends in the years between 2014 and 2020, the tendency of some of the models to warm fast, and the differences between projected and observed forcings.*

L510: this is indeed new, to me at least. Doesn't look like there was such a coherent signal in CMIP5 models (Pendergrass et al. 2017). It is also a bit counterintuitive perhaps, but definitely plausible. A little more context on the history of this (why is the result new?) and maybe maps to indicate where this is most likely to originate from would be appreciated.

*We have found some work that discusses this type of outcome in models, and traces it back to the diminished gradient from equator to poles and other aspects of the warming planet, like reduced albedo variability in high latitudes resulting from melting snow and sea ice. We plan to expand on this paragraph by citing these papers and elaborating on the origins of this phenomenon. We won't be calling it 'new' anymore. Our understanding from the literature and the fact that the signal emerges strongly after time averaging makes us doubtful that we would see a spatial signal of temperature variability straightforwardly. We would submit that a thorough investigation of the sources of this behavior is beyond our scope here, and think it warrants a study of its own, focusing on processes and dynamics (and if possible involving more models).*

L520-: This recent paper could provide some more context here: Milinski et al. (2019) The discussion section is a bit light on references to other papers. I don't want to provide a list, but I trust the authors are able to do a better job at this given their collective expertise. One specific paper that is interesting but still relatively new: Parsons et al. (2020)

*Thank you, we will add these references and rally the co-authors to provide more.*

---

## Author Comment (AC2) · 2 Dec 2020

**Response to Referee #2 (in blue italics)**

The authors present the results of CMIP6 ScenarioMIP focusing on surface air temperature and precipitation. They find similar characteristics in the temperature and precipitation patterns to those simulated by CMIP5 models. They also depict the temperature and precipitation changes under the new scenarios in CMIP6, such as SSP1-1.9, SSP3-7.0 and SSP5-3.4OS.

This is a comprehensive study of CMIP6 ScenarioMIP, summarizing the results of multiple global warming scenarios from CMIP6 and providing some new insights. However, I feel some clarifications are needed to justify the results.

*Thank you for your positive assessment and your help in clarifying our results.*

Lines 212-213 "This suggests that the model response uncertainty increases for stronger responses". Why does model response uncertainty increase for stronger responses?

*We will elaborate in the text, by pointing out that models' structural differences are encapsulated in their different climate sensitivities, and stronger forcings "exercise" that aspect of models more than weak forcings do. So, it is expected that for higher scenarios and later periods when climate sensitivity more strongly shapes the response, that response will be most variable among models that differ significantly in their climate sensitivities.*

Line 223: The authors define "separation", and use "a 21-year running mean" (Line 227) and "choose 0.1C as the threshold" (Line 228). Why is a 21-year running mean used here instead of a 9-year or 11-year running mean? The latter two running means I can understand aim to minimize the interannual variability, but what is the purpose of the 21-year running mean? Will the results of "separation" be sensitive to the two parameters of the running mean and 0.1C?

*We used 21-year running means to more conservatively beat down internal variability, noting that it is not uncommon to find the 20 (or 21 for symmetry purposes)-yr span used as a smoothing option when looking at climate outcomes for average temperature and precipitation. Undoubtedly, changing to a shorter time-mean or asking for a larger separation than 0.1C would change the results and we now add this caveat explicitly.*

Fig. 1. Are they annual mean time series of temperature and precipitation?

*Yes they are, apologies for forgetting to specify that. We actually propose to redo this piece of the analysis by using decadal running averages, also in the definition of the standard deviation, to put less focus on year-to-year variations and concentrate on climate behavior.*

Part 3.1.3 Would be it possible to show the difference between the SSPs and RCPs more explicitly? For example, plot the time series of the differences of temperature and precipitation between the SSPs and RCPs? Also show the spatial pattern of the difference, i.e., the global map of the difference of temperature and precipitation change/trend in the 21st century between the SSPs and RCPs? Besides, please explain

why are SSPs induced changes different from their corresponding RCPs? What
is the factor accounting for the difference and through what processes?
Additionally, please note the superscripts and subscripts in the denotations. For example,
"2.6Wm-2 " (Line 364), "0.5 Wm-2" (Line 400), "CO2, CH4 and N2O" (Line 419).

*We corrected subscripts and superscripts, thank you for noticing that.*
*We describe how the CMIP6 ensemble has models with higher climate sensitivity and, in the text
and Figure A7, we provide a description of the differences in radiative forcings between RCPs
and SSPs and cite references for that. We think these are the main sources of differences in the
global time series and we would rather not show year by year differences in time series between
experiments where both the forcings and the models have changed. We also point at the results
in Figure 1 of  https://agupubs.onlinelibrary.wiley.com/doi/full/10.1029/2019MS001940
attesting to substantial differences in GHG forcings between the RCPs and the SSPs. Since the
models may be modeling aerosols differently in CMIP6 (vs CMIP5) a straightforward
comparison between the two scenarios/experiment results is not possible. We added some
additional discussion and references in the text motivated by your inquiry, for example to Smith
et al. 2020 (https://doi.org/10.5194/acp-20-9591-2020) discussing differences among models in
radiative forcings and climate sensitivities and to Nicholls et al. 2020
(https://doi.org/10.5194/gmd-13-5175-2020) which evaluates differences between RCPs and
SSPs with Reduced Complexity Models. As for differences in patterns, also in response to
Reviewer 1's suggestions, we plan to show differences in the normalized patterns between
CMIP5 and CMIP6, after aggregating over scenarios (as we do not find scenarios to be a strong
source of variation). We hope this will be satisfactory.*

---

## Author Response (AR1)

We thank the editor and the two referees for their careful reading and thoughtful feedback on our paper. We have strived to take into account all the requests and suggestions made, as we detail in the point-by-point responses below, which depart only in the details – and we trust in the direction of better addressing the reviews -- from the preliminary plans we posted on the ESDD site.

The manuscript has been revised in many details since we have added the latest simulations available from the ESGF in order to fully recognize the effort of as many modeling centers participating as possible. Thus, you will find that many (if not all!) numbers have changed, but we haven't found that any of our discussion points and conclusions are affected by these changes. By using the version with track changes, that also retains the original figures, you can have an accurate assessment of this.

We have also expanded on the constrained projections and comparison between CMIP5 and CMIP6 thanks to the input from Dr. Tokarska, who ran her constrained projections on the CMIP5 ensemble, enabling an apple-to-apple comparison. Dr. Tokarska is now part of the author team.

We hope you find that we have addressed all your concerns to your satisfaction, and we look forward to the next steps. We would like to ask you for special consideration of the timeline of this next phase. In no way we give for granted that this next round will be fully satisfactory. However, **if** that turned out to be the case (which of course we hope!), we would be extremely grateful if the process could meet the IPCC AR6 WG1 deadline of January 31[st], 2021 for literature to be cited.

Thank you and happy new year,

Claudia Tebaldi and many more.

**Response to Referee #1 (in blue italics)**

This paper provides a first overview over temperature and precipitation projections from the ScenarioMIP simulations conducted for CMIP6. As such, it is more of a documentation than a cutting-edge research paper, in part also because many of the CMIP6 results are very similar to CMIP5. However, I think this is ok given the clearly defined scope of the paper and the assumed goal to support the AR6 process.

*Thank you, indeed that is the nature of this overview paper, which also serves the purpose of involving and acknowledging the representatives of the modeling centers that participated in our MIP.*

The paper is well written with clear figures and sound methods.

*Thank you, we plan to ptovide all the scripts to reproduce the analysis and figures of the paper.*

I have a few minor comments that could be considered before publication.
Minor comments:
L130-: is there a reason these papers aren't cited in the classic style with "XYZ et al."?

*This was an oversight at least in part. A couple of links have remained as such, because they are pointers to the ESGF inputs4mip downloadable data. The links to papers have been changed to standard citation style. Thank you for pointing this out.*

L147-: nothing wrong with this paragraph given the high-level nature of the study, but I think it would still benefit the reader to end the introduction with a set of specific research questions that are being addressed in this paper.

*We agree and we have added a last paragraph in the style suggested.*
*"We address questions regarding the strength of the signal under the different CMIP6 scenarios and compared to similar CMIP5 scenarios; the identification of the time of separation between the temperature trajectories under the different scenarios, and the time at which they cross global warming thresholds. We also analyze spatial patterns of change addressing questions of robustness between the CMIP5 and CMIP6 multi-model ensembles, and within the CMIP6 ensemble among models and scenarios."*

L205: "likely conservative". I think I know what the authors are trying to say, but I still think it would be clearer to be explicit and say "underestimated" and maybe also briefly explain why.

*Agreed. We have rephrased: "When considering the shaded envelopes around the ensemble mean trajectories, about 0.6°C at the lower end and 1.6°C at the upper end are added to this range. This range can be seen as reflecting the compound effects of model-response uncertainty and some measure of internal variability in the individual model trajectories, but the latter is likely underestimated, given that we are using only one run per model. The use of initial condition ensembles for each of the models would better characterize their respective internal variability (Lehner et al., 2020)."*

Paragraph on L223: Possibly relevant paper, although conceptually similar to Tebaldi and Friedlingstein: Marotzke (2019).

*Thank you, we have added this citation.*

Generally, figure labels are too small. White space between panels could be decreased to make maps bigger and better readable.

*We have redrawn all figures updating the content as a few more models have contributed their output to the ESGF in the intervening time since the paper was submitted attempting to decrease the blank space and increase the font of the labels.*

Fig A5 is insightful and given its relatively detailed discussion in the text, I suggest to

add it to Fig 2.

*It was actually like that in a previous version and we have recreated the layout by having a Figure 2a (previous Figure 2) and a Figure 2b (previous Figure A5).*

L325: ok, but how about a simple difference maps of CMIP6-CMIP5?

*Good point. We have added difference maps as a second row of panels in Figure 3 and we comment in the text on the muted features in the maps and the small areas that stand out: "We deem a rigorous quantification of the differences between patterns beyond the scope of this paper, and focus on a qualitative assessment of the similarities that surface by showing on the bottom row of Figure 3 the difference between CMIP6 and CMIP5 normalized patterns, confirming the small magnitude of the discrepancies in TAS over all regions, except for the Arctic, known to be affected by large variations among model, scenarios and internal noise. Similarly for percent precipitation the regions that stand out where the largest differences are found are the tropics, known to be affected by large variability and uncertainties."*

Fig 4b: could the Tokarska constraint be applied to CMIP5 as well to show that it makes less of a difference in CMIP5 (as expected)?

*We have been remiss in a thorough discussion of this result and we have expanded significantly this part of the analysis. Katarzyna Tokarska is now a co-author and has provided the result of applying the constraints also to CMIP5, placing the corresponding brackets on the right-hand plot of Figure 4. In the text we discuss the effect of doing so, noting how the effects of the constraints are stronger for the CMIP6 ensemble than for the CMIP5. We also included an additional figure in the appendix, Figure A6, with a starker side-by-side comparison. The new results of the constrained projections are also included in Table A6.*

L465: Semantics, but technically if the reference period is 1995-2014 and thus its mid year is 2004, then the models have 16 years, not 5, to reach 1.5C.

*Good point. After reading that sentence again we decided to limit the parenthetical to pointing out the last year of the historical period, in order to avoid complications. We have however added some more caveats and discussion of these results, pointing at the possible discrepancies between observed and modeled trends in the years between 2014 and 2020, the tendency of some of the models to warm fast, and the differences between projected and observed forcings. As an aside, we have decided to use a common subset of models for the Tier 1 experiments' crossing times (Table 1), to enhance comparability of threshold crossing among scenarios, but we have retained a table that shows results using all models available, as in the original submitted version, in the Appendix (Table A7).*

L510: this is indeed new, to me at least. Doesn't look like there was such a coherent signal in CMIP5 models (Pendergrass et al. 2017). It is also a bit counterintuitive perhaps, but definitely plausible. A little more context on the history of this (why is the result new?) and maybe maps to indicate where this is most likely to originate from would be appreciated.

*We have found some work that discusses this type of outcome in models and traces it back to the diminished gradient from equator to poles and other aspects of the warming planet, like reduced albedo variability in high latitudes resulting from melting snow and sea ice. We have cited these papers and elaborated on the possible origins of this phenomenon. We won't be calling it 'new' anymore. Our understanding from the literature and the fact that the signal emerges strongly after time averaging makes us doubtful that we would see a spatial signal of temperature variability straightforwardly. We would submit that a thorough investigation of the sources of this behavior is beyond our scope here, and think it warrants a study of its own, focusing on processes and dynamics (and, if possible, involving more models).*

L520-: This recent paper could provide some more context here: Milinski et al. (2019)
The discussion section is a bit light on references to other papers. I don't want to provide a list, but I trust the authors are able to do a better job at this given their collective expertise. One specific paper that is interesting but still relatively new: Parsons et al. (2020)

*Thank you, we have added these references and more.*

**Response to Referee #2 (in blue italics)**

The authors present the results of CMIP6 ScenarioMIP focusing on surface air temperature and precipitation. They find similar characteristics in the temperature and precipitation patterns to those simulated by CMIP5 models. They also depict the temperature and precipitation changes under the new scenarios in CMIP6, such as SSP1-1.9, SSP3-7.0 and SSP5-3.4OS.
This is a comprehensive study of CMIP6 ScenarioMIP, summarizing the results of multiple global warming scenarios from CMIP6 and providing some new insights. However, I feel some clarifications are needed to justify the results.

*Thank you for your positive assessment and your help in clarifying our results.*

Lines 212-213 "This suggests that the model response uncertainty increases for stronger responses". Why does model response uncertainty increase for stronger responses?

*We have elaborated in the text, by pointing out:*
*"This suggests that the model response uncertainty increases for stronger responses, an expected result as in higher scenarios and later periods climate sensitivity – which significantly differs among the models -- more strongly influences the model response (Lehner et al. 2020)."*

Line 223: The authors define "separation", and use "a 21-year running mean" (Line 227) and "choose 0.1C as the threshold" (Line 228). Why is a 21-year running mean

used here instead of a 9-year or 11-year running mean? The latter two running means I can understand aim to minimize the interannual variability, but what is the purpose of the 21-year running mean? Will the results of "separation" be sensitive to the two parameters of the running mean and 0.1C?

*We have added more discussion and justification of these two choices, with better anchoring (we believe) to previous work. We have also explicitly mentioned that the precise estimates reported would change if those two choices (running means and 0.1C as the threshold) were changed, but We hope that the new explanation makes them seem less arbitrary. The introductory text to the analysis of separation now reads:*
*"In order to characterize when pairs of scenarios diverge, we define separation the first occurrence of a positive difference between two time series, one under the higher and one under the lower forcing scenarios, which is then maintained for the remainder of the century. This is similar to Tebaldi and Friedlingstein (2013, TF13 in the following), which used the first occurrence of a significant trend in the year-by-year differences, then justified by the RCPs under consideration, among which only the lowest, RCP2.6, flattened out over the century. In that case, the remainder of the RCPs considered followed an increasing trajectory, with differential rates of increase, therefore justifying the expectation that year-by-year differences would eventually show a significant and persisting trends. Among the new scenarios at least two are expected to follow a flat trajectory, or even a slight peak and decline (SSP1-1.9 and SSP1-2.6) rendering the expectation of a trend in their differences untenable. We therefore adopt a slightly different definition, here, and we also note that this definition would need to be modified if overshoot scenarios -- crossing their reference as they decrease -- were the main focus of this analysis. Also, this is not the only way to define separating scenarios and other studies have applied different, but still fairly similar, definitions, e.g., recently, Marotzke (2019). We use time series of GSAT after applying a 21-year running mean, as we are concerned with differences in climate rather than in individual years, whose temperatures are affected by large variability (this is the part of the definition that takes the place of the consideration of long-term trends in TF13. We also need to choose a threshold by which we deem the difference "positive" and somewhat discernible (this takes the place of asking for a significant trend in TF13). To do so, we use the results in Tebaldi et al., 2015, where the regional sensitivities of temperature and precipitation to changes in global average temperature were quantified. According to that analysis, 0.1°C of difference in 20-yr means of GSAT was the lowest value at which a multi-model ensemble consistently had a positive fraction of the grid-cells experiencing significant warming."*

Fig. 1. Are they annual mean time series of temperature and precipitation?

*Yes they were, apologies for forgetting to specify that. However, we have decided to redo all analysis involving time series using a 21-yr running mean, to focus attention on climatic time scales rather than year-to-year variability. The trajectories in the plots showing time series are now much smoother across the board.*

Part 3.1.3 Would be it possible to show the difference between the SSPs and RCPs more explicitly? For example, plot the time series of the differences of temperature and precipitation between the SSPs and RCPs? Also show the spatial pattern of the difference, i.e., the global map of the difference of temperature and precipitation

change/trend in the 21st century between the SSPs and RCPs? Besides, please explain why are SSPs induced changes different from their corresponding RCPs? What is the factor accounting for the difference and through what processes? Additionally, please note the superscripts and subscripts in the denotations. For example, "2.6Wm-2 " (Line 364), "0.5 Wm-2" (Line 400), "CO2, CH4 and N2O" (Line 419).

*We corrected subscripts and superscripts, thank you for noticing that.*
*We describe how the CMIP6 ensemble has models with higher climate sensitivity and, in the text and Figure A7, we provide a description of the differences in radiative forcings between RCPs and SSPs and cite references for that. We think these are the main sources of differences in the global time series and we would rather not show year by year differences in time series between experiments where both the forcings and the models have changed. We also point at the results in Figure 1 of  https://agupubs.onlinelibrary.wiley.com/doi/full/10.1029/2019MS001940 attesting to substantial differences in GHG forcings between the RCPs and the SSPs. Since the models may be modeling aerosols differently in CMIP6 (vs CMIP5) a straightforward comparison between the two scenarios/experiment results is not possible. We added some additional discussion and references in the text motivated by your inquiry, for example to Smith et al. 2020 (https://doi.org/10.5194/acp-20-9591-2020) discussing differences among models in radiative forcings and climate sensitivities and to Nicholls et al. 2020 (https://doi.org/10.5194/gmd-13-5175-2020) which evaluates differences between RCPs and SSPs with Reduced Complexity Models. As for differences in patterns, also in response to Reviewer 1's suggestions, we now show differences in the normalized patterns between CMIP5 and CMIP6 in Figure 3, after aggregating over scenarios (as we do not find scenarios to be a strong source of variation). We hope this will be satisfactory.*

---

## Author Response (AR2)

Dear Dr. Liu,

First and foremost, thank you for handling our manuscript revision so swiftly, and thank you to the two reviewers for the fast turn-around, especially during these times. It is greatly appreciated.

We are happy that the reviewers find our responses satisfactory. We are also happy to try and address the last two points raised by Reviewer 1. We have attempted to do so navigating the line between adding value to the discussion of our analysis, as the reviewer has helped with, and avoiding speculating on the sources of the behaviors under consideration, given the limits set for our overview study, which the reviewer acknowledges and agrees with.

We have modified the two parts of the text where the two issues are treated in the following manner (the new wording is highlighted in the text pasted below), using the reviewer suggestions, also including the Yip et al., 2011 reference in our list. We also included a new sentence in the Discussion section with regard to the results in precipitation variability, and two additional references, Pendergrass et al., 2017 and Yun et al., 2020. We hope this satisfies these last concerns.

Please note a correction we performed of an oversight (due to my submitting the paper at a time when the second author, responsible for the data processing was already on vacation). The time series of global quantities shown in Figures 1, 4, 6 and 7, and A2, were smoothed by an 11-yr running mean, not a 21-yr running mean. Of course, that is not changing any results.

Thank you once again for your attention and commitment to this process.
All the best – and happy new year,

Claudia Tebaldi

**About pattern differences:**
We deem a rigorous quantification of the differences between patterns beyond the scope of this paper, and focus on a qualitative assessment of the similarities that surface by showing on the bottom row of Figure 3 the difference between CMIP6 and CMIP5 normalized patterns, confirming the small magnitude of the discrepancies in TAS over all regions, except for the Arctic, known to be affected by large variations among model, scenarios (with a possible role of the lowest scenario in CMIP6, SSP1-1.9, whose land-sea ratio has likely no equivalent among the CMIP5 scenarios, but further, more rigorous investigation is needed to confirm this) and internal noise (likely playing a minor role given the number of model and scenarios contributing to these averages). Similarly for percent precipitation the regions that stand out where the largest differences are found are the tropics, known to be affected by large variability and uncertainties. In this case the possible role of aerosol forcing (Yip et al., 2011) warrants further investigation, especially as we consider that SSP3-7.0 forcing composition and trajectory are quite different from previous scenarios'. As mentioned, the use of these experiments in conjunction with their variants by LUMIP and AerChemMIP could further attribute some of these scenario-dependent features to differences in regional forcing like land-use or aerosols. Also, a subset of CMIP6 models are running the CMIP5 RCPs, and results from those experiments will allow a clean analysis of variance, partitioning sources between model and scenario generations.

**About changes in ensemble spread over time:**

A deeper investigation of the sources of changes in variability for both variables (which could also tackle how much of the changes in precipitation variability is directly connected to that of GSAT, and what other sources may be at play) is beyond our scope but will be facilitated by the availability of these CMIP6 IC ensembles in addition to the already well studied CMIP5-era large IC ensembles (Deser et al., 2020).

**More related to that in the Discussion section:**

The decadal scale results appear at odds with recent studies that detected increased variability of precipitation with warming (Pendergrass et al., 2017; Yun et al., 2020), and call for in-depth studies of the sources and robustness of the behavior here described.

**New References:**

Pendergrass, A.G., Knutti, R., Lehner, F., Deser, C., and Sanderson, B.M.: Precipitation variability increases in a warmer climate. Scientific Reports, 7, 17966, https://doi.org/10.1038/s41598-017-17966-y, 2017.

Yip, S., Ferro, C.A.T., Stephenson, D.B., and Hawkins, E.: A simple, coherent framework for partitioning uncertainty in climate predictions. Journal of Climate, 24, 17, 4634-4643, https://doi.org/10.1175/2011JCLI4085.1, 2011.

Yun, K.-S., Lee, J.-Y., Timmermann, A., Stein, K., Stuecker, M.F., Fyfe, J.C., and Chung, E.S.: Increasing ENSO-rainfall variability due to changes in future tropical temperature-rainfall relationship. Nature Communications Earth and Environment, in press.